Corrected: Author correction

# Lung endothelial cell antigen cross-presentation to CD8+T cells drives malaria-associated lung injury

Carla Claser [1,5], Samantha Yee Teng Nguee[1,2,5], Akhila Balachander [1], Shanshan Wu Howland [1], Etienne Becht[1], Bavani Gunasegaran [1], Siddesh V. Hartimath [3], Audrey W.Q. Lee [1], Jacqueline Theng Theng Ho[1], Chee Bing Ong [4], Evan W. Newell[1], Julian Goggi [3], Lai Guan Ng [1] & Laurent Renia [1,2]

Malaria-associated acute respiratory distress syndrome (ARDS) and acute lung injury (ALI) are life-threatening manifestations of severe malaria infections. The pathogenic mechanisms that lead to respiratory complications, such as vascular leakage, remain unclear. Here, we confirm that depleting CD8+T cells with anti-CD8β antibodies in C57BL/6 mice infected with *P. berghei* ANKA (PbA) prevent pulmonary vascular leakage. When we transfer activated parasite-specific CD8+T cells into PbA-infected TCRβ$^{-/-}$ mice (devoid of all T-cell populations), pulmonary vascular leakage recapitulates. Additionally, we demonstrate that PbA-infected erythrocyte accumulation leads to lung endothelial cell cross-presentation of parasite antigen to CD8+T cells in an IFNγ−dependent manner. In conclusion, pulmonary vascular damage in ALI is a consequence of IFNγ-activated lung endothelial cells capturing, processing, and cross-presenting malaria parasite antigen to specific CD8+T cells induced during infection. The mechanistic understanding of the immunopathogenesis in malaria-associated ARDS and ALI provide the basis for development of adjunct treatments.

[1] Singapore Immunology Network (SIgN), A*STAR, 8A Biomedical Grove, Level 3 & 4 Immunos Building, Singapore 138648, Singapore. [2] Department of Microbiology and Immunology, Yong Loo Lin School of Medicine, National University of Singapore, 5 Science Drive 2 Blk MD4, Level 3, Singapore 117545, Singapore. [3] Isotopic Molecular Imaging Laboratory, Singapore Bioimaging Consortium (SBIC), A*STAR, 11 Biopolis Way, #02-02 Helios, Singapore 138667, Singapore. [4] Histolopathology/Advanced Molecular Pathology Lab, Institute of Molecular and Cell Biology (IMCB), A*STAR, 61 Biopolis Drive, Level 6 Proteos Building, Singapore 138673, Singapore. [5] These authors contributed equally: Carla Claser, Samantha Yee Teng Nguee. Correspondence and requests for materials should be addressed to C.C. (email: carla_claser@immunol.a-star.edu.sg) or to L.R. (email: renia_laurent@immunol.a-star.edu.sg)

Malaria-associated acute respiratory distress syndrome (ARDS) and acute lung injury (ALI) are severe and life-threatening manifestations of *Plasmodium* infection. ARDS affect 5–25% adults infected with *Plasmodium falciparum*[1,2]. However, a recent increase in the incidence of malaria-associated ARDS resulting from two other *Plasmodium* species has been reported, *P. vivax* in Southeast Asia and South America[3] and, *P. knowlesi*, a zoonotic parasite also prevalent in Southeast Asia[4]. Both pathologies develops mostly in non-immune adults and pregnant women with placental malaria[1,2]. ARDS, the severe form of ALI, is characterized by alveolar inflammation, alveolar-capillary membrane damaged and pulmonary edema[1,5]. Malaria-associated ARDS onset is sudden, rapid and can occur any time during malaria infection, even after anti-malarial treatment has substantially reduced parasitemia levels[2,5]. Currently, treatment for ARDS patients consists of a combination of antimalarial drugs with management therapy (e.g., ventilation support), albeit with limited success[2,5,6]. Thus there is an urgent need for new adjunct therapies.

*Plasmodium* infections induce systemic inflammation that can be amplified locally by endothelial and inflammatory cells in response to sequestered infected red blood cells (iRBC)[7]. Pro-inflammatory mediators such as TNFα and/or IFNγ increase the expression of adhesion molecules, such as ICAM-1, VCAM-1, and P-selectin[8] on the surface of endothelial cells. Indeed, electron microscopy analysis of post-mortem lung histological sections from ARDS patients[9,10] has revealed the accumulation of leukocytes (monocytes, neutrophils, macrophages, and other cell types) and iRBC, suggesting that iRBC and immune-cell sequestration may be key pathogenic factors. In addition, inflammation can increase endothelial lining permeability and leads to protein-rich plasma fluid leakage and ultimately, pulmonary edema. Edema has been observed in the alveolar airspace and lung interstitium in malaria-infected human patients suffering from ARDS[2,5]. This serious clinical problem can be life-threatening due to impaired gas exchange.

Because of the difficulty to perform time-course experiments clinically and limited access to human lung tissues, mouse malaria models have been developed using different parasite/mouse strain combinations to decipher the pathogenic mechanisms underlying ARDS. *P. berghei* NK65 (PbNK65) infection of C57BL/6 mice[11], PbA infection of DBA/2 mice[12], and PbA infection in C57BL/6 mice[13,14], elicit a lung pathology similar to human ARDS. A common finding in all these models is the presence of leukocyte infiltrate into the lungs and vascular leakage leading to edema[12–14]. Depleting these CD8+T cells partially reduced lung edema in PbA-infected C57BL/6 mice[15] and in PbNK65-infected C57BL/6 mice[11]. Here, we investigate the function of parasite-specific CD8+T cells and the pathogenic mechanisms resulting in ALI in a PbA-induced malaria mouse model. In addition, we provide evidence that lung endothelial cells are able to cross-present parasite antigens to specific CD8+T cells causing lung injury.

## Results

### PbA*luc*-infected mice exhibit lung edema and vascular leakage.
Here, we have used an established rodent model of malaria infection where majority (>60–80%) of susceptible C57BL/6 mice infected with a transgenic luciferase-expressing line derived from *P. berghei* ANKA (PbA*luc*), develop a lethal neurological disease during the first 2 weeks of infection (Fig. 1a, b). Mice that do not develop neurological signs were referred as non-cerebral malaria (NCM), succumbing to death due to high parasitemia and anemia. In this model of experimental cerebral malaria (ECM), all mice also develop an underlying acute lung pathology,

characterized by a significant increase in lung wet to dry weight ratio by 7 days post infection (dpi), indicative of edema (Fig. 1c). Since in malaria patients with ALI, edema arises from lost integrity of the alveolar capillary barrier and increased alveolar permeability[5], we thus quantified disruption of the alveolar-capillary barrier by in vivo (Supplementary Fig. 1A) or ex vivo lung imaging (Supplementary Fig. 1B, C) using Tracer-653. This tracer is a fluorescent dye that distributes passively in the blood and allows in vivo measurement of vascular permeability[16]. Vascular leakage in the lung region was first detected at 6 dpi in PbA*luc*-infected C57BL/6 mice, and further increased 7 dpi compared to 3–5 dpi (Fig. 1d). This was further confirmed when leakage was measured in isolated lungs removed at 7 dpi (Supplementary Fig. 1B, C). This increase in vascular leakage did not correlate with the amount of parasites sequestered in the lungs, as detected by bioluminescence imaging[17], because iRBC accumulation reached a plateau 5 dpi when leakage was yet to be observed (Fig. 1e). Interestingly, vascular leakage occurred independently of the presence of neurological signs (Fig. 1f). We also used another non-invasive technique, in vivo magnetic resonance imaging (MRI), to detect edema in PbA*luc*-infected mice. We were able to detect a significant increase in the water content in the lungs of infected mice at 6 and 7 dpi (Fig. 1g), confirming and validating the tracer findings. Altogether, these data suggest that the vascular leakage is not due to accumulation of iRBC alone in the lungs and may be a consequence of the local and systemic inflammatory responses induced by the parasites.

### Activated CD8+T cells migrate to the lungs during infection.
Leukocytes have been shown to migrate to the lung microvasculature and interstitium during ALI[13,18]. To thoroughly define the leukocyte populations infiltrating the lungs, we performed a comprehensive flow cytometry phenotyping of the immune cells present in the lungs at 7 dpi (Fig. 2a and Supplementary Table 1). There was no significant difference in total leukocyte number, and of the myeloid subsets, only monocyte-derived dendritic cells (DC) and monocyte-derived macrophages were increased in the lungs of PbA*luc*-infected mice compared to naive (non-infected) mice. Neutrophils, plasmacytoid DC (pDC) and conventional DC (cDC) subsets were all decreased in number in PbA*luc*-infected mice. Of the lymphocyte subsets, only the CD8+T cell subset increased in number and in percentage as compared to naïve mice. We also observed that total lung CD8+T cells had an activated phenotype (as defined by the expression of lymphocyte function-associated antigen 1 (LFA-1)[19] (Fig. 2b).

Using tetramers specific for the highly immunogenic PbA Pb1 epitope[20], we observed a significant increase in parasite-specific CD8+T cells in the lungs (Fig. 2c) that expressed high levels of IFN-γ and GrzB (Fig. 2d). Moreover, these activated CD8+T cells were found in the spleen and lungs as early as 5 dpi (Supplementary Fig. 2A).

We next used cytometry by time-of-flight (CyTOF) to perform deep phenotyping of the CD8+T cells isolated from naive and infected spleens (where the CD8+T cells are initially primed) at 5 dpi and in the lungs at 7 dpi. Following CyTOF with 39 markers, dimensionality reduction using UMAP revealed that the phenotypes of the CD8+T cells present in the lungs differed from those present in the spleen (Supplementary Fig. 2B). Lung CD8+T cells form one homogeneous population (Fig. 3a, b), displaying a fully activated phenotype evidenced by effector markers (CD44+ CD62L−), and up-regulation of markers such as ICOS, KLRG1, Ki67, and GITR (Fig. 3c). Taken together, these data suggest that for vascular leakage to occur, leukocytes that migrate to the lungs have to be fully mature and functional.

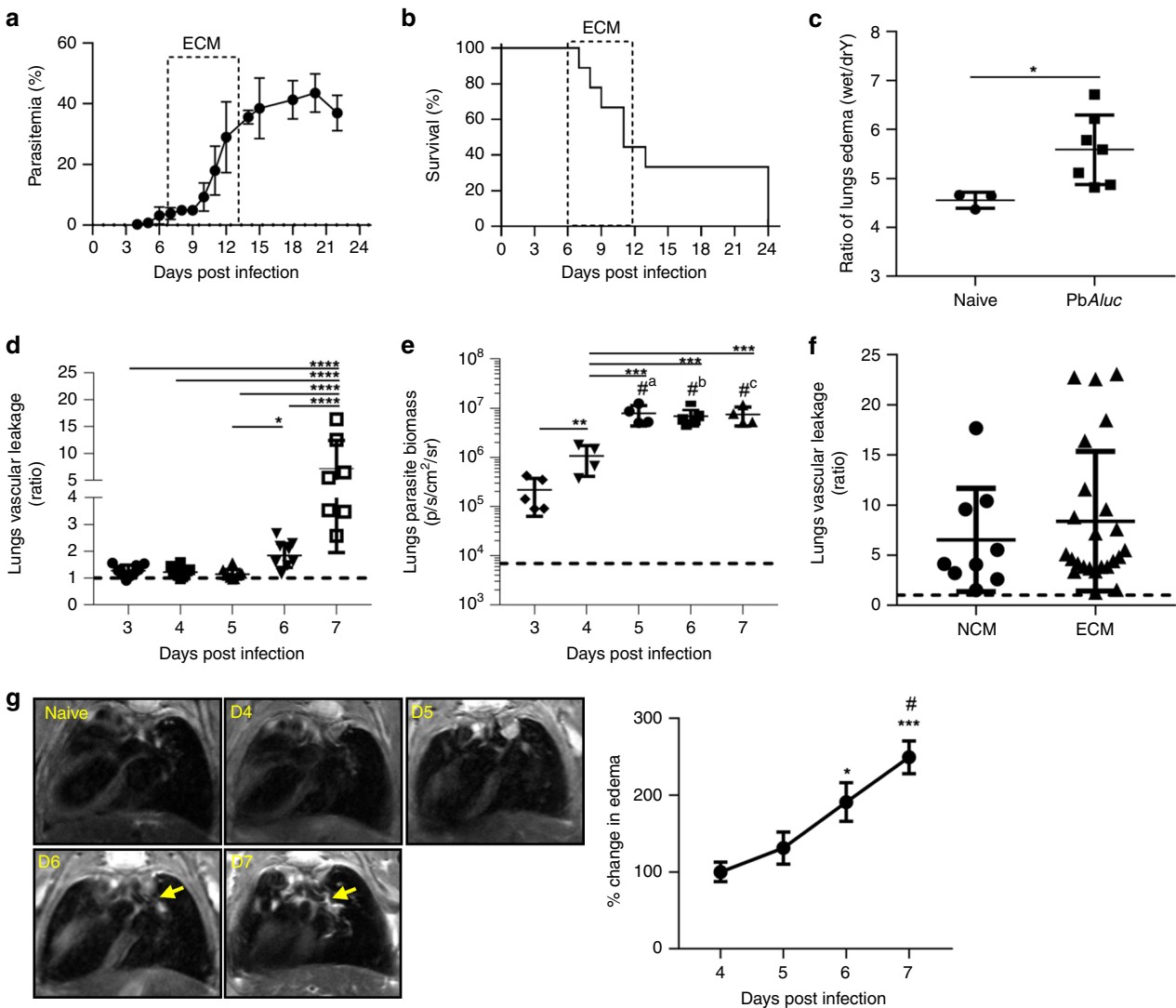

**Fig. 1** Pathophysiological parameters measured in the lungs during PbA*luc*-infection in C57BL/6 mice. **a** Peripheral parasitemia levels and **b** survival curve of PbA*luc*-infected mice ($n = 9$). The boxed area indicates the experimental cerebral malaria (ECM) phase. **c** Ratio of lung edema (wet-to-dry weight ratio) at 7 days post infection (dpi) in infected ($n = 7$) and naive C57BL/6 ($n = 3$) mice. **d** Ratio of in vivo lung vascular leakage measured by Tracer-653 dye at 3–7 dpi ($n = 10$). **e** Ex vivo quantification of parasite biomass in the lungs based on luciferase activity of luciferase-tagged parasite after perfusion at 3 ($n = 5$) 4 ($n = 4$), 5 ($n = 4$), 6 ($n = 5$) and 7 dpi ($n = 4$). The data shown are representative of two independent experiments. **f** Ratio of in vivo lung vascular leakage assessed in mice with non-cerebral malaria (NCM) ($n = 9$) and with experimental cerebral malaria (ECM) ($n = 24$). The black dashed line at $y = 1$ in (**d**–**f**) represents the ratio of the tracer reading from naïve C57BL/6 mice ($n = 3$). **g** MRI images of naïve C57BL/6 ($n = 4$) and PbA*luc*-infected mice ($n = 5$) taken consecutively from 4–7 dpi. Yellow arrows show edema (left hand side images). Edema quantification based on manual segmentation/ volume calculation (right hand side graph). The data represent the mean ± SD; *$p < 0.05$ by Mann–Whitney test (**c**); *$p < 0.05$, **$p < 0.01$, ***$p < 0.001$, ****$p < 0.0001$ by ANOVA with Bonferroni's post-test (**d**–**e**, **g**). Figure **e**, "#" represents statistical significance compared to 3 dpi; [a,b,c]$p < 0.0001$; Figure **g**, represents statistical significance compared to 4 dpi ($p < 0.05$); "#" represents statistical significance compared to 5 dpi (***$p < 0.001$)

**Anti-CD8β antibody treatment ameliorates pulmonary damages.** Anti-CD8 antibody treatment may prevent or reduce lung pathology, by inhibiting edema in *P. berghei*-infected mice[11,15,21]. We confirmed this finding in PbA*luc*-infected C57BL/6 mice. A single injection of anti-CD8β, but not control antibody,antibody at 6 dpi, when CD8+T cells were already in the lung tissue and vascular leakage was observable, protected the mice from ECM development prolonging their survival (Supplementary Fig. 3B–C). Antibody depletion had no effect on parasitemia (Fig. 4a) but led to significant decrease in pulmonary vascular leakage at 7 dpi (Fig. 4b). We also found that CD8+T cell depletion decreased the PbA*luc* parasite density in the lungs at 7 dpi, as determined using ex vivo bioluminescence imaging (Fig. 4c). CD8+T cell depletion did not prevent or reduce the migration of other

immune-cell population to the lungs (Supplementary Table 4). This strongly suggests a direct role for CD8+T cells rather than an indirect one through the recruitment of other effector cells.

Histologic analysis of PbA*luc*-infected lungs from control group at 7 dpi revealed that mice exhibited severe intra-alveolar edema, intra-alveolar erythrocyte extravasation, and thickening of the alveolar septa characterized by increased eosinophilic fibrillar matrix and loss of wall integrity (Fig. 4d, middle panel). Macrophages/monocytes were found in the alveoli and contained cytoplasmic granular dark-brown malaria pigment (hemozoin) (Fig. 4d, middle panel). By contrast, PbA*luc*-infected mice treated with anti-CD8β antibodies exhibited only mild histologic changes, with less severe pulmonary edema and minimal intra-alveolar hemorrhage compared to untreated and naïve mice

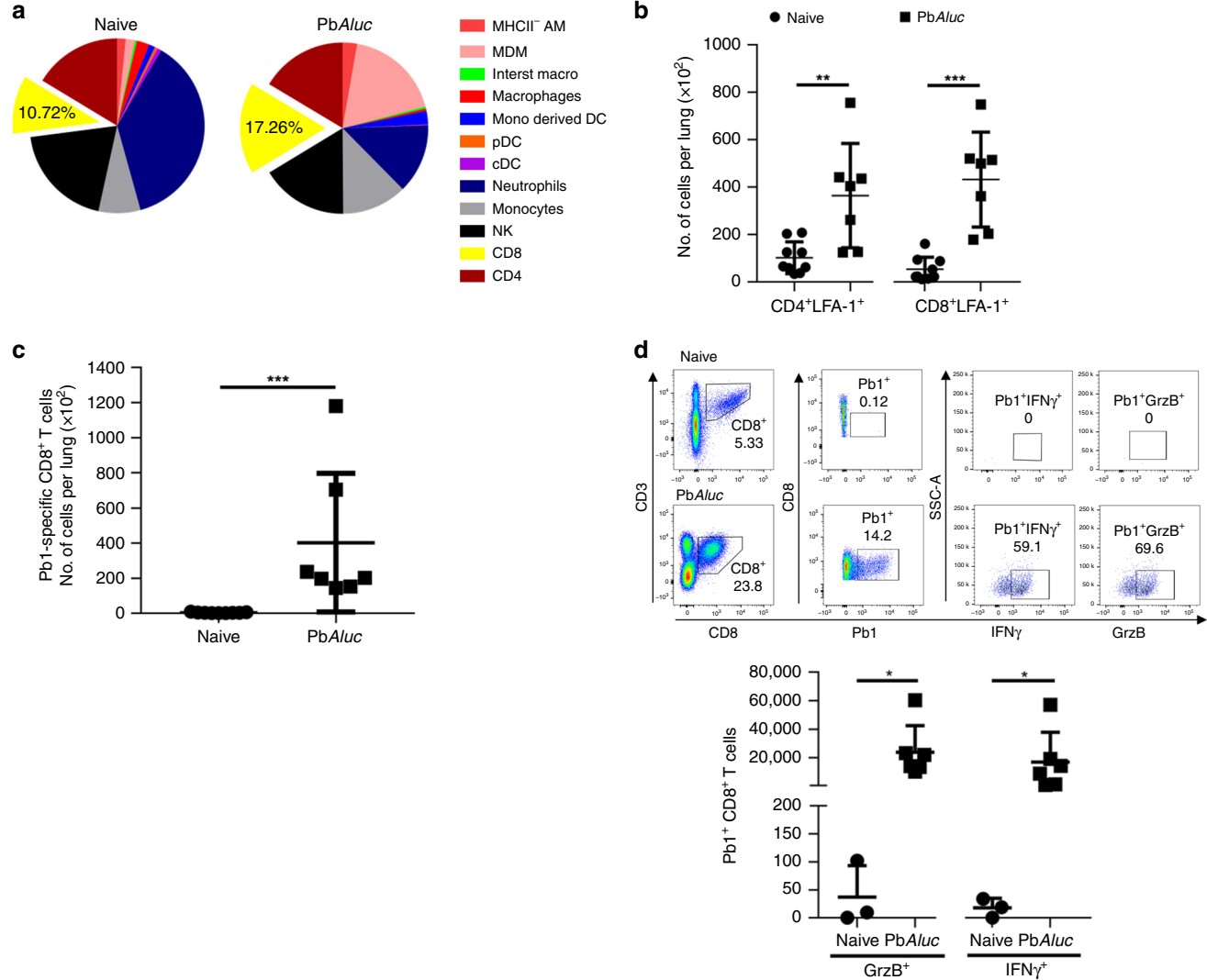

**Fig. 2** Inflammatory response induced in the lungs during PbAluc-infection in C57BL/6 mice. **a** Relative percentage of the mean counts of each immune-cell population found in the lungs of naive ($n = 9$) and PbAluc-infected mice ($n = 11$) at 7 dpi derived from the total number of cells shown in Supplementary Fig. 1. MHCII⁻ AM MHC-Class II negative alveolar macrophages, MDM monocyte-derived macrophages, DC dendritic cell, cDC conventional dendritic cell, pDC plasmacytoid DC, NK natural killer. **b** Total number of CD4⁺LFA-1⁺ and CD8⁺LFA-1⁺ T cells and **c** Pb1-specific CD8⁺T cells accumulated in the lungs of naive ($n = 9$) and infected mice ($n = 7$). **d** Tetramer and intracellular cytokine staining was performed on lungs sequestered leukocytes from naive ($n = 3$) and PbAluc-infected mice ($n = 6$). Cells were incubated with Brefeldin for 3 h without re-stimulation. The indicated dot plot (top) was used to analyze the Pb1-specific CD8⁺T cells and to gate the IFN-γ⁺ and GrzB⁺ population. The total number of Pb1-specific CD8⁺T cells secreting each cytokine is represented graphically (bottom). The data shown are representative of two independent experiments. The data represent the mean ± SD; *$p < 0.05$, **$p < 0.01$, ***$p < 0.001$ by Mann–Whitney test (**b–d**)

(Fig. 4d, right panel). When comparing the pathological scores, we found that untreated PbAluc-infected mice had a higher total score than treated PbAluc-infected mice (Fig. 4e). Quantification of the pathological scores confirmed that anti-CD8β antibody treatment significantly diminished pulmonary edema (Fig. 4f) and intra-alveolar erythrocyte extravasation (Fig. 4g) when compared to untreated PbAluc-infected mice.

Because we observed substantial pulmonary vascular leakage in PbAluc-infected mice, we next sought to investigate whether the damage extended to the pulmonary epithelium and if so, whether CD8⁺T-cell depletion could also prevent damage to the epithelium wall. Here, we performed immunofluorescence on paraffin-embedded lung sections using antibodies against E-cadherin and *zonula occludens* 1 (ZO-1) proteins to identify epithelial cells and tight junctions, respectively (Fig. 4h). Based on the average intensity of ZO-1 signal, we found that ALI was also

associated with loss of epithelial intercellular junctions, defined by decrease in ZO-1 signal intensity in PbAluc-infected mice. The ZO-1 signal intensity was almost restored to a similar level as naïve (uninfected and untreated) mice after anti-CD8β antibody treatment (Fig. 4i). In summary, these data suggest that CD8⁺T cells contribute to pulmonary endothelial and epithelial damage and that this effect may be prevented or halted by anti-CD8 antibody treatment.

**CD8⁺T-cell adoptive transfer model recapitulates lung injury.** To further validate the pathogenic role of CD8⁺T cells in inducing pulmonary injury, we established cell adoptive transfer experiments. Specifically, we transferred CD8⁺T cells from PbAluc-infected (7 dpi) or naive (uninfected) C57BL/6 mice into TCRβ⁻/⁻ mice (devoid of T cells) that were infected 3 days prior

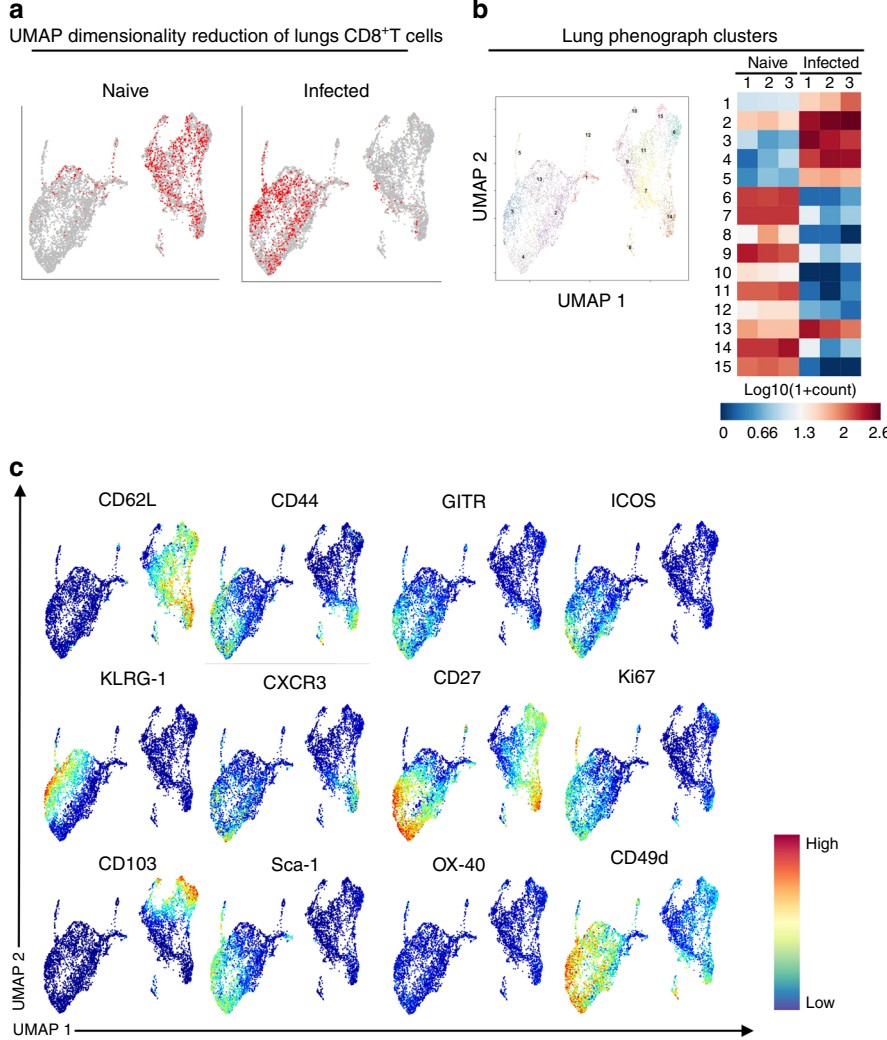

**Fig. 3** UMAP dimensionality reduction of CD8+T cells. **a** UMAP dimensionality reduction of lung CD8+T cells isolated from naive (*n* = 3) and PbA*luc*-infected (*n* = 3) mice at 7 dpi was color-coded (red dots represents where the CD8+T cells are located). A representative plot of three mice is shown. **b** Lung CD8+T cell clusters segregation by UMAP identified a total of 15 different clusters. Median expression of lung phonograph clusters across individual samples plotted and summarized as a heat map. **c** UMAP dimensionality reduction of lung CD8+T cells color-coded by relative intensity in each channel. The figure shows only the highly expressed markers

with PbA*luc*. We then used in vivo Tracer imaging after 4 days to measure pulmonary vascular leakage. The PbA*luc*-infected TCRβ$^{-/-}$ mice that received CD8+T cells from PbA*luc*-infected donor mice exhibited more pulmonary vascular leakage than control infected TCRβ$^{-/-}$ and infected TCRβ$^{-/-}$ mice receiving CD8+T cells from naive mice (Fig. 5a, right panel).

To confirm that parasite-specific CD8+T cells were responsible for the pathological effect observed, we adoptively transferred purified CD8+T cells isolated from BSL8.4 TCR transgenic mice (where 98% of CD8+T cells express a TCR specific for the Pb1 epitope) into TCRβ$^{-/-}$ mice, and again we observed increased vascular leakage (Fig. 5a, right panel). Adoptive transfer of CD8+T cells isolated from infected C57BL/6 donors did not induce vascular leakage in naïve TCRβ$^{-/-}$ hosts (Fig. 5a, right panel), ruling out a non-specific effect of parasite-specific CD8+T cells. Interestingly, the number of activated and specific CD8+T cells that migrated into the lungs was much lower in naïve versus infected TCRβ$^{-/-}$ hosts (Supplementary Fig. 5A), further ruling out non-specific effects of the transferred CD8+T cells.

We next visualized CD8+T cell migration and retention in the lungs using tissue clearing and light sheet microscopy which allowed 3D imaging of the lung tissues. Here, we adoptively transferred enriched CD8+T cells from μGFP mice (these cells express GFP, allowing in vivo and ex vivo tracking of donor leukocytes) into PbA*luc*-infected TCRβ$^{-/-}$ mice at 3 dpi. A high number of CD8+T cells was observed throughout the PbA*luc*-infected lungs at 7 dpi (Fig. 5b, Supplementary Movies 1–2). Interestingly, CD31 (endothelial) staining revealed a significant decrease in the average compactness of the lung tissue in PbA*luc*-infected mice compared to naïve mice (Fig. 5c).

Taken together, all these data highlight the importance of parasite accumulation in the lung, because pulmonary vascular leakage occurred when CD8+T cells isolated from PbA*luc*-infected donor mice were transferred into infected but not naïve TCRβ$^{-/-}$ mice.

**Lung endothelial cells cross-present PbA antigen to CD8+.** It has been postulated that lung endothelial cells can also function as non-professional antigen-presenting cells, by expressing major histocompatibility complex (MHC) class I (MHC I) or class II (MHC II) molecules in response to certain inflammatory cytokines (e.g., IFN-γ) that are triggered during infection[22,23]. We

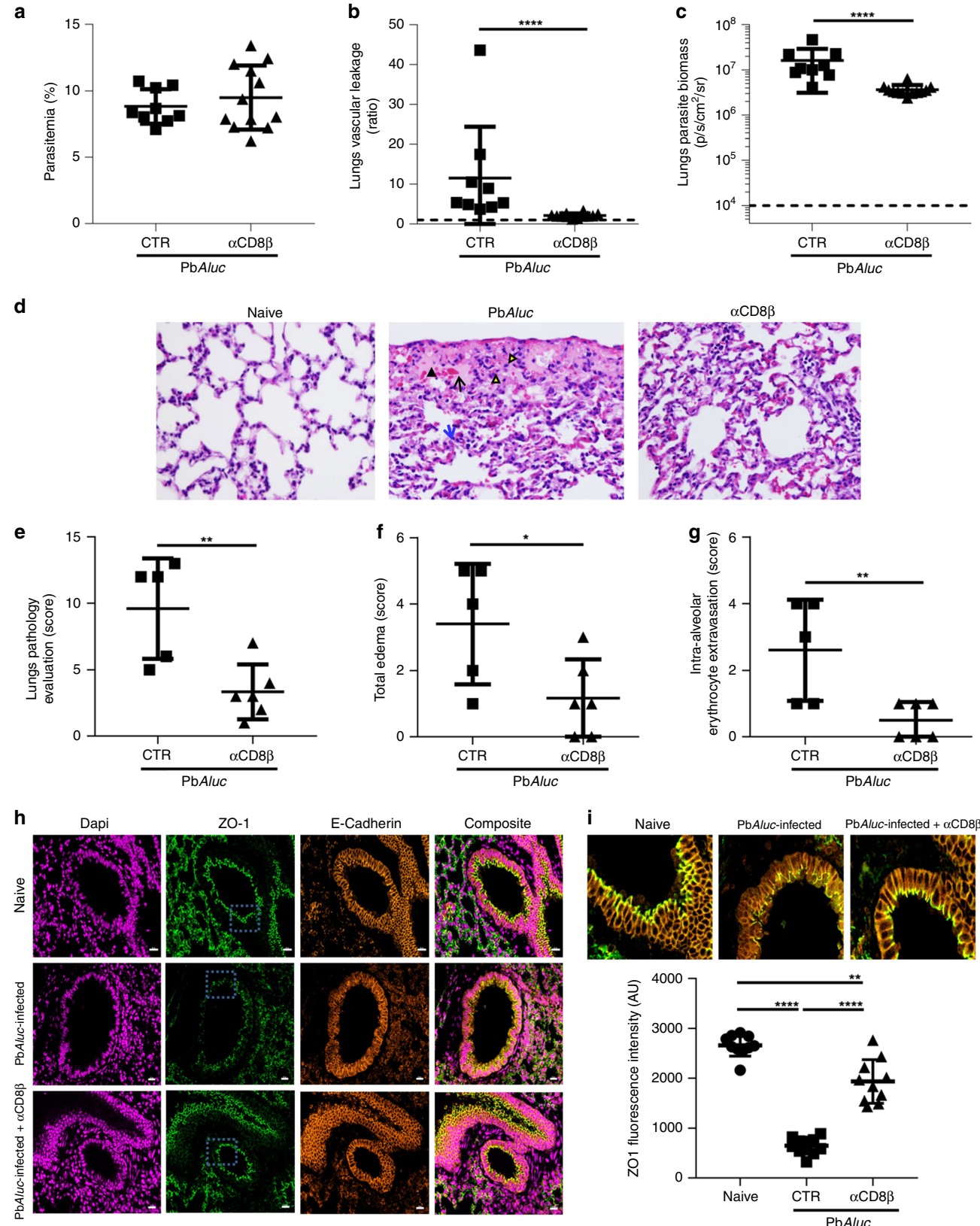

have previously shown that during ECM, brain endothelial cells from PbA*luc*-infected mice uptake and process parasite antigens and cross-present parasite-derived epitopes to CD8+T cells, resulting in blood–brain barrier damage[20,24]. Here we adapted our brain endothelial cross-presentation assay[20] to determine whether lung endothelial cells have the same capability. We

sorted CD45−CD31+ lung endothelial cells and CD45+ cells (both infiltrating and resident hematopoietic cells) from PbA*luc*-infected C57BL/6 mice and incubated them individually with a reporter cell line (LR-BSL8.4a) harboring a Pb1-specific TCR (13). Indeed, we observed that lung endothelial cells but not CD45+ cells, captured parasite antigen during the course of

**Fig. 4** Anti-CD8β protects PbA*luc*-infected mice from pulmonary vascular leakage. Comparison between PbA*luc*-infected mice (CTR) and PbA*luc*-infected mice depleted of CD8⁺ T cells (αCD8β) at 6 dpi. **a** Peripheral parasitemia of CTR (given rat IgG1 isotype control) ($n = 9$) and αCD8β ($n = 12$) mice at 7 dpi. **b** Ratio of in vivo lung vascular leakage measured by Tracer-653 dye and **c** ex vivo quantification of parasite biomass in perfused lungs based on the luciferase activity of the PbA*luc* parasite in CTR ($n = 9$) and αCD8β ($n = 12$) mice at 7 dpi. The data shown was pooled from two independent experiments. **d** H&E staining of histologic lung sections. Black arrowhead = intra-alveolar edema; black arrow = intra-alveolar erythrocyte extravasation; blue arrow = thickened alveolar septa; and yellow arrowheads = intra-alveolar macrophages/monocytes. Pathological changes were graded on a scale of 1 to 5 (0 = no abnormalities; 5 = severe): **e** overall lung pathology score; **f** total edema and **g** severity of intra-alveolar erythrocyte extravasation in non-infected/naïve ($n = 3$), PbA*luc*-infected CTR ($n = 5$) and αCD8β ($n = 6$) mice. Naïve mice were not depicted in the graphs as they obtained a pathological score of zero. The black dashed line at $y = 1$ in (**b**) represents the ratio of the tracer reading from naïve C57BL/6 mice ($n = 3$). For ex vivo quantification of parasite biomass in the lungs, data are expressed on a log scale, with naïve C57BL/6 mice indicated by the black dashed line in (**c**). **h** Immunofluorescence of FFPE lung sections of naive, PbA-infected and PbA-infected plus αCD8β-treated mice at 7 dpi, stained with ZO-1 and E-cadherin antibodies. Images are representative of two mice per group. **i** Quantification of the average pixel intensity of ZO-1 drawn over the apical ZO-1 signal (blue box in (**h**)). Scale bar = 20 mm. The quantification shown is representative of three independent experiments. The data represent the means ± SD; *$p < 0.05$, **$p < 0.01$, ****$p < 0.0001$, by Mann–Whitney test (**a–c**), unpaired student's *t* test (**e–g**) or by ANOVA with Bonferroni's post-test (**h–i**)

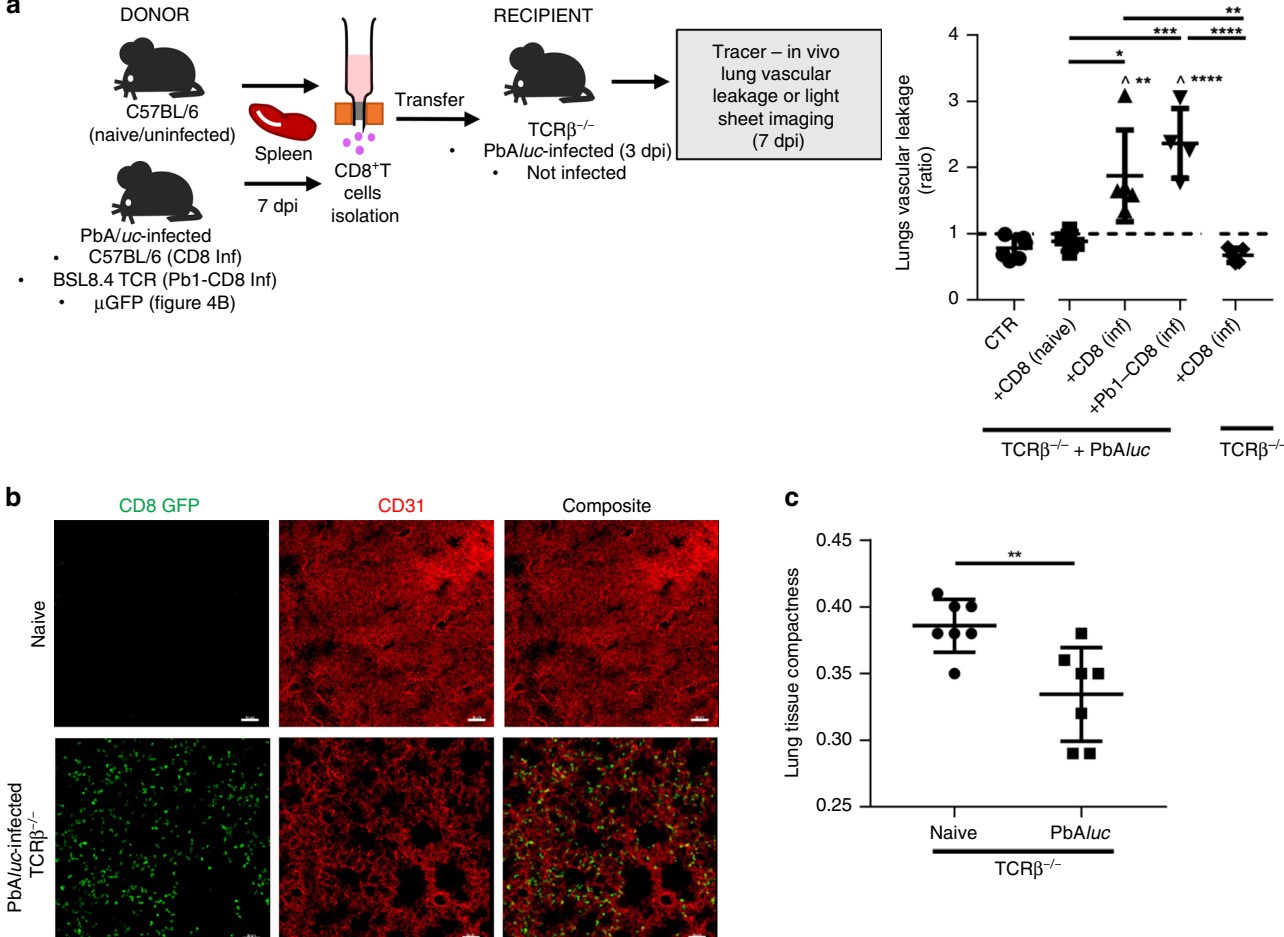

**Fig. 5** CD8⁺T cells migrate in the lungs of PbA*luc*-infected TCRβ⁻/⁻ and is associated with pulmonary vascular damage. **a** Adoptive transfer protocol of CD8⁺T cells. Splenocytes were isolated from PbA*luc*-infected C57BL/6 mice at 7 dpi and CD8⁺T cells were isolated and enriched by negative selection using magnetic bead-based cell sorting. Purified CD8⁺T cells (with at least 95% purity) were transferred to PbA*luc*-infected TCRβ⁻/⁻ mice on 3 dpi. Pulmonary vascular leakage was assessed 4 days post CD8⁺T cell transfer. The ratio of in vivo lungs vascular leakage was measured by Tracer-653 dye in PbA*luc*-infected TCRβ⁻/⁻ (CTR, did not receive CD8⁺T cells) mice ($n = 6$); PbA*luc*-infected TCRβ⁻/⁻ mice that were adoptively transferred with enriched CD8⁺T cells isolated from the spleen of naïve C57BL/6 (+CD8 (naïve)) ($n = 4$), 7 dpi PbA*luc*-infected C57BL/6 (+CD8 (inf)) mice ($n = 5$), 7 dpi PbA*luc*-infected BSL8.4 TCR transgenic mice (+Pb1-CD8 (inf)) ($n = 4$); and naïve TCRβ⁻/⁻ that received CD8⁺T cells from 7 dpi PbA*luc*-infected C57BL/6 mice ($n = 5$). The data represent the mean ± SD; *$p < 0.05$, **$p < 0.01$, ***$p < 0.001$, ****$p < 0.0001$, ˆ statistically significant compared to CTR ($p < 0.05$) by ANOVA with Bonferroni's post-test. The black dashed line at $y = 1$ in (**a**) represents the ratio of tracer reading from naïve mice ($n = 3$). **b** Confocal imaging of cleared lungs of naïve and PbA*luc*-infected TCRβ⁻/⁻ mice that have received CD8⁺T cells isolated from μGFP mice at 7 dpi and stained with anti-GFP, which indicates CD8⁺T cells (green). Scale bar = 50 mm. **c** Lung tissue compactness was quantified as a ratio of volume of CD31 mask (from (**b**)) to the total volume of selected region of interest. The data are representative of two mice and represent the mean ± SD; **$p < 0.01$ by unpaired student's *t*-test

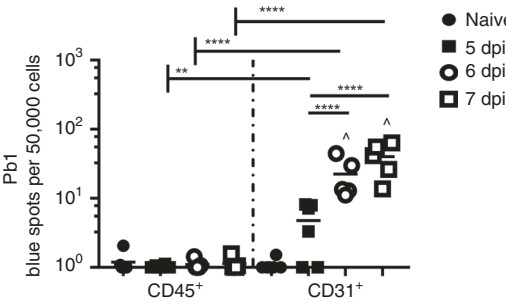

**Fig. 6** Lung endothelial cell cross-presentation of PbAluc antigen to Pb1-specific CD8+T cells during PbAluc infection. Lung microvessels were isolated from naive (n = 6) and PbAluc-infected C57BL/6 5 (n = 6), 6 (n = 5) and 7 dpi (n = 5), and sorted into two populations: CD45+CD31− leukocytes and CD45-CD31+ endothelial cells. Each sorted population was tested for Pb1 cross-presentation by incubating with LR-BSL8.4a cells and staining with X-gal. The spot count was normalized per 50,000 cells and is represented on a log scale. The data represent the mean; **p < 0.01, ****p < 0.0001, ˆ statistically significant compared to naive mice (p < 0.05) by ANOVA with Bonferroni's post-test

infection and cross-presented it in a time-dependent manner, from as early as 6 dpi (Fig. 6). These data suggest that endothelial cells cross-presenting parasite antigens are the target of parasite-specific CD8+T cells and are essential for lung vascular leakage to occur.

**IFN-γ is involved in lung injury and cross presentation.** Malaria infection induces a systemic pro-inflammatory response, in part mediated by the cytokine IFN-γ[25,26]. We previously observed that PbAluc-infected IFN-γ−/− mice are completely resistant to ECM[17]. IFN-γ has a central role in brain endothelial cell activation, as shown in vivo by increased expression of adhesion molecules such as ICAM-1 and VCAM-1[27] and leukocyte recruitment in the brain[28,29] during PbA infection. We thus asked whether IFN-γ influences leukocyte migration and endothelial cell activation in the lungs. We infected IFN-γ−/− mice with PbAluc and although there was no significant difference in parasitemia at 7 dpi (Fig. 7a), we detected lower parasite density in the lungs compared to PbAluc-infected WT C57BL/6 mice (Fig. 7b). Pulmonary vascular leakage was also abrogated in infected IFN-γ−/− mice (Fig. 7c). The lack of IFN-γ, however, resulted in an increased total number of leukocytes in the lungs (Supplementary Fig. 6A), with higher numbers of total and activated CD8+T cells (Fig. 7d, e) and Pb1-specific CD8+T cells (Fig. 7f) compared to infected WT mice. These data were surprising since we have demonstrated that activated and parasite-specific CD8+T cells (as shown above) were essential for ALI to develop. We then shifted our focus from the effector cells to the suspected target cells, the lung endothelial cells, as previous studies have demonstrated that IFN-γ is essential for cross-presentation by brain endothelial cells[30]. The level of lung endothelial cell cross-presentation from infected-IFN-γ−/− mice was much lower than the observed levels in infected WT C57BL/6 mice (Fig. 7g). IFN-γ stimulates lung endothelial cells to up-regulate antigen-presenting MHC class I molecules[31]. To determine if this was also the case during PbA infection, we infected WT and IFN-γ−/− mice with PbAluc and phenotyped the antigen-presenting H-2Db (the MHC class I molecule presenting the Pb1 epitope) (Fig. 7h) and H2-Ab (MHC class II) molecules (Supplementary Fig. 6D). We found that these molecules were all significantly up-regulated on lung endothelial cells of WT but not IFN-γ−/− mice during PbA infection. Taken together, these data show that IFNγ is essential for cross-presentation of parasite

antigens by the lung endothelial cells and confirms that this mechanism is critical for the development of ALI.

**Lung endothelial cells requires IFNγ to cross-present PbA antigen.** To gain insights into how lung endothelial cells cross-present PbAluc during ALI, we sought to recapitulate the process in vitro using primary lung endothelial cell cultures. Cultured CD31+CD45− lung endothelial cells expressed the endothelial markers VE-Cadherin (CD144), CD62L and CD34 (endothelial progenitor markers) (Fig. 8a). To retain the original phenotype, we passaged the primary cultures only once before performing in vitro cross-presentation assays. Un-stimulated and IFNγ-stimulated lung endothelial cells were incubated with PbAluc mature iRBCs for 24 h, washed and then co-incubated with LR-BSL8.4a reporter cells to detect parasite-derived Pb1 cross-presentation. Only lung endothelial cells that had been exposed to both IFN-γ and iRBCs from PbA-infected mice cross-presented the Pb1 epitope (Fig. 8b). This ability of lung endothelial cells to cross-present antigens only after IFN-γ stimulation corroborates with our ex vivo findings that cross-presentation does not occur in IFN-γ−/− mice.

**Parasite load determines lung endothelial cells cross-presentation.** Thus far, we have demonstrated that iRBCs are sequestrated in the lungs and presented by lung endothelial cells. We next investigated whether intensive anti-malarial drug treatment could prevent ALI by reducing the parasite density in the lungs and cross-presentation by lung endothelial cells. To this aim, we treated PbAluc-infected C57BL/6 mice with a combined artesunate and chloroquine drug treatment, for 48 h starting on 5 dpi when malaria infection was firmly established, mimicking clinical situations where anti-malarial treatment is frequently administered only after parasitemia is detected microscopically. Peripheral parasitemia levels (Fig. 9a) and parasite biomass in the lungs (Fig. 9b) dropped significantly to a barely detectable level after antimalarial treatment. This drop correlated with virtually-complete protection from pulmonary vascular leakage (Fig. 9c). Total activated and Pb1-specific CD8+T cell numbers significantly increased in the lungs of mice that received the treatment at 5 dpi (Fig. 9d–f), as part of a phenomenon of overall increased migration of leukocytes (Supplementary Fig. 7A–C). After treatment, the reduced amount of parasite antigen available in the lungs, led to a significant decrease in Pb1 cross-presentation by lung endothelial cells down to background levels (Fig. 9g). Taken together, these data further demonstrate that ALI results from cross-presentation of malaria parasite antigens by lung endothelial cells, a mechanism that is dependent on parasite load in the lung and that can be prevented by aggressive anti-malarial treatment.

**Discussion**

This study has demonstrated that cross-presentation of PbA malaria antigen by IFN-γ-activated lung endothelial cells to parasite-specific CD8+T cells induces lung damage (vascular leakage) in the PbA-infected C57BL/6 ALI/ARDS mouse model.

Previous histology studies have revealed the presence of inflammatory cells and iRBC in the lungs of patients or mice with ALI and/or ARDS, suggesting that these cells or the inflammation that they may induce, may have an important role in malaria lung pathogenesis[9]. Using the fluorescent dye Tracer-653, we were able to monitor vascular leakage, a hallmark of ALI, ex vivo but also in vivo in a non-invasive and dynamic-manner. This approach helped us demonstrate that vascular leakage is not solely dependent on parasitemia and parasite sequestered in the lungs, since the latter were already at the maximal saturation at 5 dpi

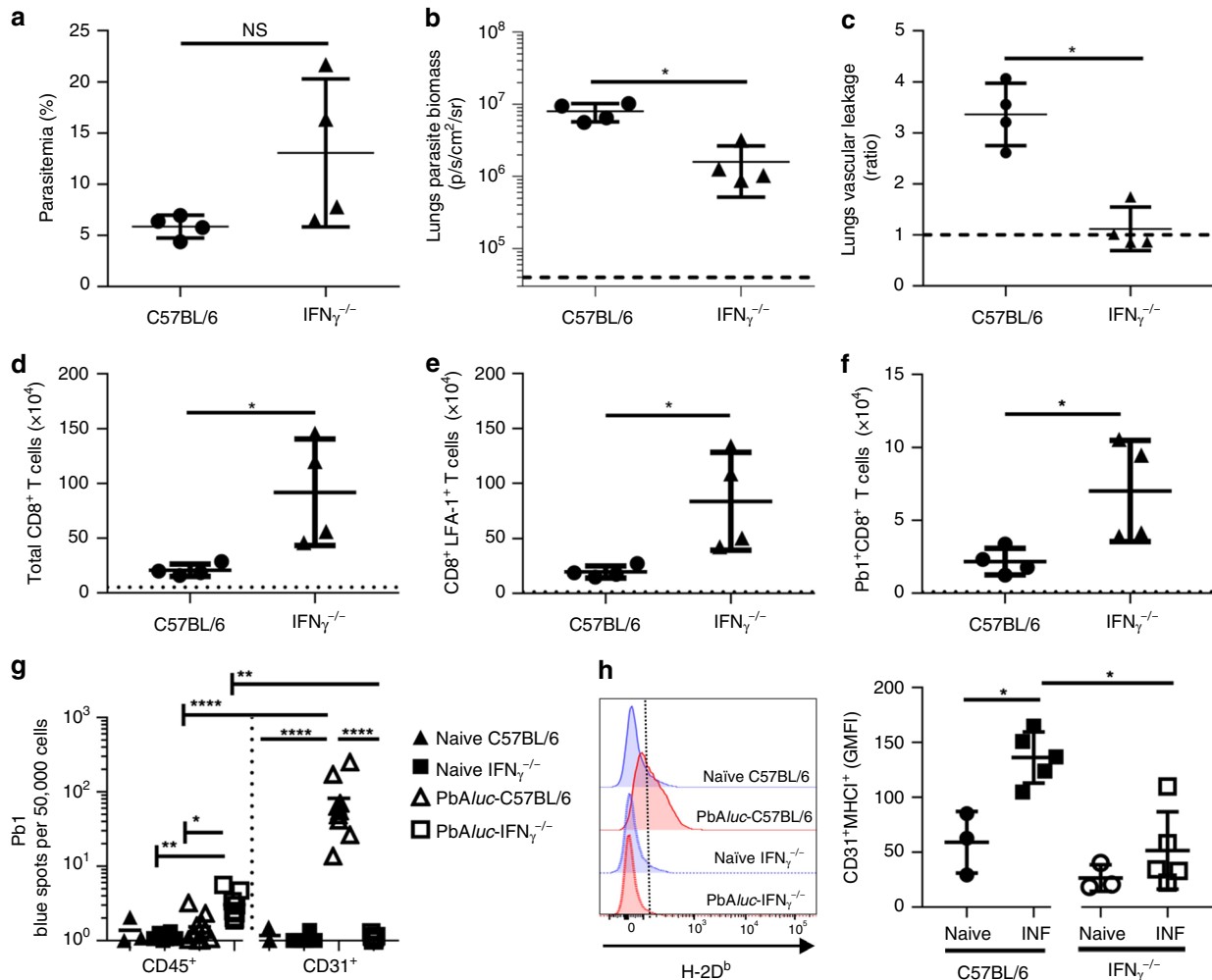

**Fig. 7** Absence of IFN-γ prevents lung injury and hinders cross-presentation of malaria antigens by lungs microvessels. **a** Peripheral parasitemia and **b** ex vivo quantification of parasite biomass in the lungs based on luciferase activity after perfusion, of PbAluc-infected C57BL/6 ($n = 4$) and IFN-γ$^{-/-}$ ($n = 4$) at 7 dpi. **c** Ratio of in vivo lung vascular leakage measured by Tracer-653 at 7 dpi ($n = 4$ in each group). **d** Total number of CD8$^+$T, **e** CD8$^+$ LFA-1$^+$ T and **f** Pb1-specific CD8$^+$T cells accumulated in the lungs at 7 dpi ($n = 4$). The data shown are representative of three independent experiments. **g** Lung microvessels were isolated from naïve C57BL/6 (N C57BL/6) ($n = 3$), naïve IFN-γ$^{-/-}$ (N IFN-γ$^{-/-}$) ($n = 4$), PbAluc-infected C57BL/6 ($n = 10$) and PbAluc-infected IFN-γ$^{-/-}$ ($n = 9$) mice at 7 dpi. Each population was tested for Pb1 cross-presentation. The spot count was normalized per 50,000 cells and represented on a log scale. **h** Representative histograms of lung endothelial cells (CD45$^-$CD31$^+$) from naïve C57BL/6 ($n = 3$) and IFN-γ$^{-/-}$ ($n = 3$) mice (blue line), and PbAluc-infected C57BL/6 ($n = 4$) and IFN-γ$^{-/-}$ ($n = 5$) mice (red line) at 7 dpi that were stained for MHC-I (H-2D$^b$). Graphs represent the geometric mean fluorescence intensities (GMFI) of MHC-Class I (H-2D$^b$) on CD31$^+$ lung endothelial cells from naïve and infected (INF) mice. For ex vivo quantification of parasite biomass in the lungs in (**b**), data are expressed on a log scale, with naive mice represented by the black dashed line. The black dashed line at $y = 1$ in (**c**) represents the ratio of tracer reading from naïve mice ($n = 3$). The black dotted line in (**d–f**) represents the value of each respective cell population from naïve mice for quantification of the immune-cell populations in the lungs. The data represent the mean ± SD; NS: not statistically Significant, *$p < 0.05$, by Mann–Whitney test (**a–f**, **h**); *$p < 0.05$, **$p < 0.01$, ****$p < 0.0001$, by ANOVA with Bonferroni's post-test (**g**)

when no vascular leakage was observed (Fig. 1d–e). Using MRI 9.4 at Tesla, we were also able to conduct a longitudinal study in mice and demonstrate that pulmonary edema occurs at 6 dpi (Fig. 1g), concurrently with vascular leakage. Of note, the only published data on MRI findings relate to patients with cerebral malaria[32,33], pediatric cerebral malaria patients[34], and ECM-susceptible and resistant mice infected with PbA[35,36]. Our data indicate that this technique may complement or replace traditional X-ray imaging of lung edema.

Our extensive phenotypic analysis of lung cell infiltrates revealed that monocyte-derived DCs, monocyte-derived macrophages (MDMs) and CD8$^+$T cells increased in number in PbA-infected mice compared to naïve mice, while pDC and cDC subsets decreased in number; no notable differences were observed in the frequencies of monocytes and neutrophils

(Supplementary Fig. 1). MDMs are activated and recruited to the lungs during PbA infection in C57BL/6 mice, actively phagocytose iRBC, thereby limiting the extent of parasite accumulation and lung injury[37]. The role of monocyte-derived DCs in the PbA-infected C57BL/6 ALI/ARDS model is unknown; however, it was recently shown that monocyte-derived DCs are recruited to the lungs of C57BL/6 mice infected with PbNK65 parasites to mediate lung injury[38]. These cells produce TNF-α and inducible nitric oxide synthase in the lungs and their migration depends on IFN-γ produced by CD8$^+$T cells[38]. Further studies are needed to determine whether these TNF-α/iNOS (TIP) DCs are also important in PbA-infected C57BL/6 ALI/ARDS model.

In our study, PbA infection did not increase neutrophil recruitment to the lungs. This finding differs from previous studies that showed an increase in neutrophils in the lung tissue and

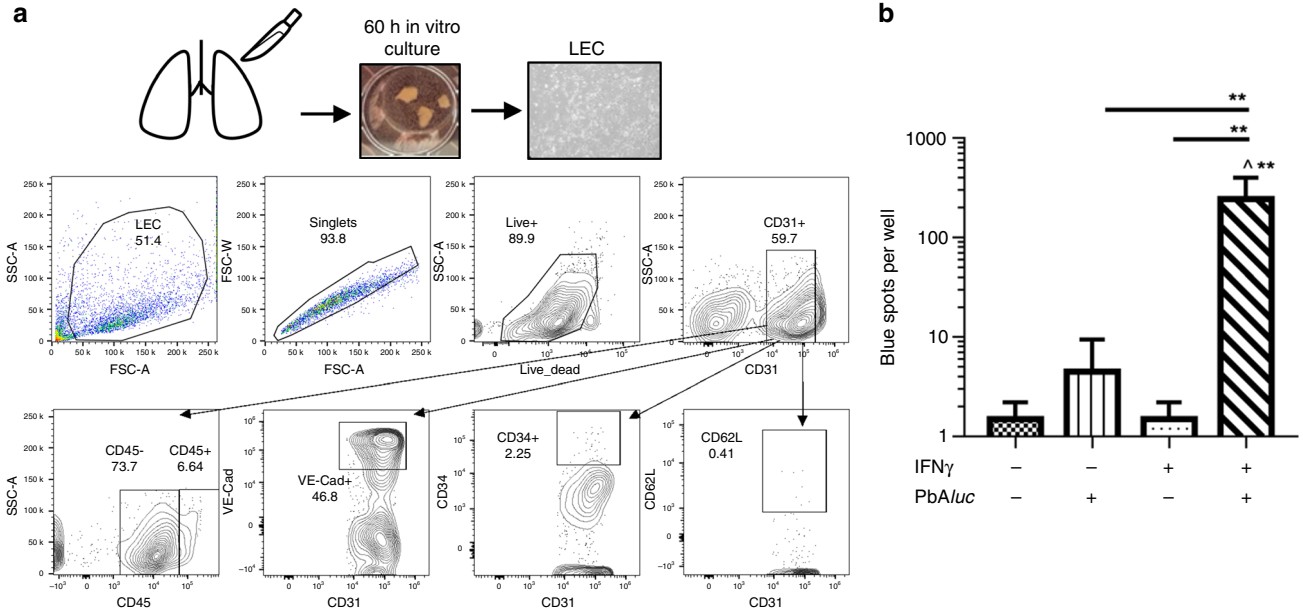

**Fig. 8** IFNγ-stimulated lung endothelial cells (LEC) cross-present in vitro PbA*luc* antigen to reporter cells. **a** LEC primary culture protocol. Mouse lungs were excised and cut into 1 × 1 × 1 mm³ pieces and placed in a 6-well plate for 60 h. The histograms are representative of two independent culture experiments and show characterization of the endothelial cells by flow cytometry (staining for CD45, CD31, VE-Cadherin (CD144), CD34 and CD62L (L-selectin)). Almost 80% of the cultured cells were CD45⁻CD31⁺ endothelial cells. **b** LEC seeded in triplicate wells of a 48-well plate were stimulated or not with 10 ng/ml IFN-γ for 24 h and then incubated or not with 5 × 10⁵ thawed PbA*luc*-matured iRBCs for a further 24 h. The wells were washed and co-cultured with LR-BSL8.4a reporter cells overnight stained for β-gal. The data represent the mean ± SD; **p < 0.01, by ANOVA with Bonferroni's post-test. ^Statistically significant compared to the first well (negative control) (p < 0.01)

bronchoalveolar lavage fluid of PbA-infected DBA2 mice[18] and PbNK65-infected C57BL/6 mice[1]. This difference is likely due to the different times at which tissues were processed for neutrophil enumeration: we quantified neutrophils 7 dpi (before mice died from ECM) while the latter studies quantified neutrophils 10–11 dpi. Interestingly, experiments using neutrophil-depleting antibodies in two previous studies suggested that neutrophils were the main effector cells[13,18]. These studies used antibodies against CD11a/LFA-1 or Gr-1 to deplete neutrophils. However, these antibodies are known to also deplete activated CD8⁺T cells[39], which we showed here to be essential for pulmonary vascular leakage to occur. Another study demonstrated that treatment with dexamethasone, an anti-inflammatory glucocorticoid, could completely abrogate ARDS development without affecting neutrophil number[11]. More work is needed, therefore, to clarify the precise role of neutrophils in malaria-induced lung injury.

Tissue accumulation of CD8⁺T cells has been previously described as a key driver of ALI/ARDS pathogenesis[1,15]. Activated CD8⁺T cells may cause damage to endothelial cells and/or alveolar epithelial cells through their cytotoxicity activity, leading to circulatory shock[11,15]. Here, we used antibody-mediated depletion and adoptive transfer experiments (TCRβ⁻/⁻ mice) to show that parasite-specific CD8⁺T cells are indeed responsible for vascular leakage and ensuing lung pathology. Depletion of CD8⁺ T cells done on 6 dpi (when the pathology is already observable) did not affect migration of other immune cells and their accumulation in the lungs of infected mice (Supplementary Table 4). This suggests that migration of other cells (monocytes, neutrophils or else) into the lungs is independent of CD8⁺T cells, and that they have no or a limited role in lung vascular leakage observed on 7 dpi (Fig. 4b). We found that total sequestered CD8⁺T cells, as well as parasite-specific CD8⁺T cells, produced IFN-γ and granzyme B. These CD8⁺T cell populations have been shown to promote rupture of the blood–brain-barrier during ECM[40,41]. We observed that the extent of pulmonary vascular

leakage was associated with disrupted tight junction (ZO-1) in the pulmonary epithelium (Fig. 4i). This finding suggests that granzyme B, and possibly perforin, may be involved in the rupture of endothelial cell lining through reduced expression of tight junction protein, as shown for the blood–brain-barrier during ECM[27,42] (Fig. 10). Defining how CD8⁺T cells alter the endothelial lining, whether is through granzyme B release, deserves further study. Interestingly, CD8⁺T cell depletion with a CD8β-specific monoclonal antibody almost fully preserved pulmonary epithelial integrity, suggesting that the damage induced by CD8⁺T cells can be prevented, as summarized in our proposed mechanism for malaria-associated ALI (Fig. 10). So far, only dexamethasone has been shown to decrease the infiltration of CD8⁺T cells in the lungs, and to protect infected mice from lung pathology[11]. This drug was used in the past in some clinical trial to improve survival in *P. falciparum* infected patients with cerebral malaria, however without success since it proved deleterious to the patients[43,44]. The usage of dexamethasone in ALI/ARDS has yet to be explored in humans.

Since we have shown that parasite-specific CD8⁺ T cells were the main effector cells, we next identified their target cells. Our study demonstrated that lung endothelial cells from infected animals were the target since they were the only cells able to cross-present parasite antigen to a reporter cell line expressing a T cell receptor specific for Pb1, an immunodominant parasite epitope (Fig. 6). We ruled out contaminating leukocytes (resident or infiltrated) as the cross-presenting population, as these cells stimulated reporter cells only minimally. The capacity of lungs endothelial cells to cross-present the malaria antigen at different time points of infection, strongly suggested that cross-presentation was the limiting factor for vascular leakage to occur, as despite the simultaneous presence of iRBC and specific-CD8⁺T cells in the lung tissue at 5 dpi, no significant difference in cross-presentation was observed compared to naive (Fig. 6).

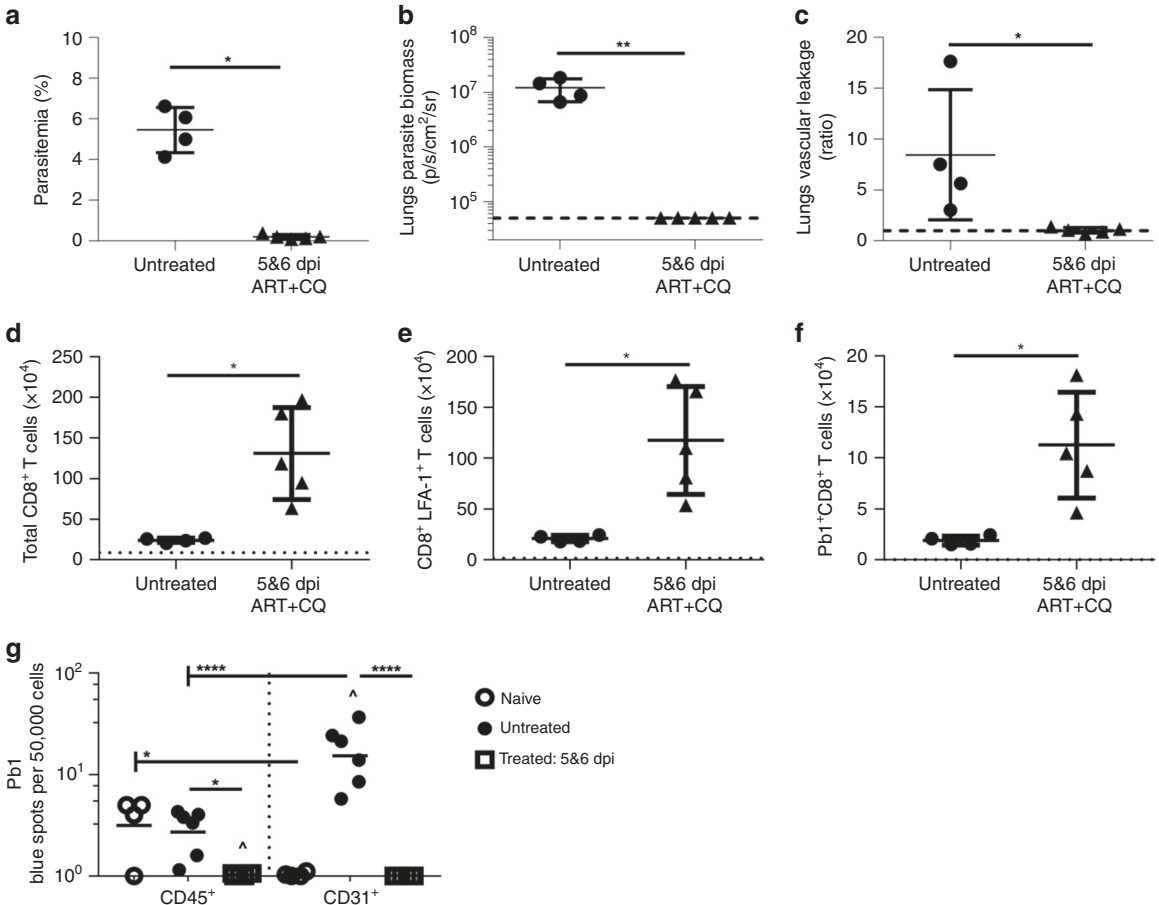

**Fig. 9** Anti-malarial treatment protects infected mice from pulmonary leakage and abrogates cross-presentation of malaria antigen by lungs endothelial cells. Comparison between PbA*luc*-infected C57BL/6 mice (untreated) ($n = 4$) and infected mice treated with artesunate and chloroquine (ART + CQ) at 5 and 6 dpi (5&6) ($n = 5$). All analyses were performed at 7 dpi. **a** Peripheral parasitemia. **b** Ex vivo quantification of parasite biomass in the lungs based on luciferase activity of PbA*luc* after perfusion. **c** Ratio of lung vascular leakage in vivo at 7 dpi measured by Tracer-653 dye. Total number of **d** CD8+T cells, **e** CD8+ LFA-1+ T and **f** Pb1-specific CD8+T cells accumulated in the lungs at 7 dpi. The data were pooled from two independent experiments. **g** Lung microvessels were isolated at 7 dpi from the naïve C57BL/6 ($n = 4$), untreated PbA*luc*-infected C57BL/6 mice ($n = 6$) and infected mice treated with ART + CQ at 5 and 6 dpi ($n = 6$) and sorted into two populations: CD45+31− leukocytes and CD45−CD31+ endothelial cells. Each sorted population was tested for Pb1 cross-presentation by incubating with LR-BSL8.4a cells and staining with X-gal. The spot count was normalized per 50,000 cells and represented on a log scale. For ex vivo quantification of parasite biomass in the lungs in (**b**), the data were expressed on a log scale, with naïve C57BL/6 mice represented by a black dashed line. The black dashed line at $y = 1$ in (**c**) represents the ratio of tracer reading from naïve C57BL/6 mice ($n = 3$). The black dotted line in (**d–f**) represents the value of each respective cell population from naïve C57BL/6 mice for quantification of immune-cell populations in the lungs. The data represent the mean ± SD; *$p < 0.05$, **$p < 0.001$, by Mann Whitney test (**a–f**); *$p < 0.05$, ****$p < 0.0001$, ^ statistically significant compared to naive ($p < 0.001$, $p < 0.0001$), by ANOVA with Bonferroni's post-test (**g**).

Indeed, in-depth profiling of CD8+T cells by CyTOF found that CD8+T cells present in the lungs at 7 dpi differ from those present in the spleen at 5 dpi. While CD8+T cells present in the lungs 7 dpi displayed effector status and had down–regulated CD62L, those in the spleen 5 dpi still expressed CD62L but not effector markers (Fig. 3). We hypothesize, therefore, that the cells present in the lungs 5 dpi share a similar phenotype to those in the spleen 5 dpi (Fig. 6 and Supplementary Fig. 2A). A kinetic of malaria-specific CD8+T cells at the early stage of PbA infection, as well as their expression profile, deserves further characterization.

Malaria infection is associated with increased production of pro-inflammatory cytokines, such as IFN-γ. This cytokine has a potent effect on the endothelium, leading to increased expression of adhesion molecules and chemokine receptors, parasite adhesion and immune-cell trafficking[26,45,46] (Fig. 10). IFNγ-stimulation also enhances the antigen processing capability of endothelial cells[20,47]. Here we demonstrated that in the absence of

IFNγ, there was a significant decrease in parasite sequestration in the lungs (Fig. 7b) despite no difference in peripheral parasitemia (Fig. 7a). IFN-γ was also required, in vivo and in vitro, for lung endothelial cells to cross-present the malaria antigen to the reporter cells (Figs. 7g and 8b). In the absence of IFN-γ, MHC class I expression on lung endothelial cells was low (Fig. 7h), which could explain, in part, the impaired cross-presentation. Interestingly, the increased accumulation of CD8+T cells in PbA*luc*-infected IFN-γ−/− mice suggested that CD8+T-cell activation and migration to the lung, as described in other studies[48], can occur independently of IFN-γ-signaling. Most importantly, our overall findings show that IFN-γ has a vital role in cross-presentation by lung endothelial cells and reinforced its critical role for ALI development, regardless of CD8+T cell accumulation (Fig. 10).

In our mouse model, lung vascular leakage occurred regardless of the development of neurological signs (Fig. 1f), which is also characterized by brain vascular leakage[15]. This result is surprising

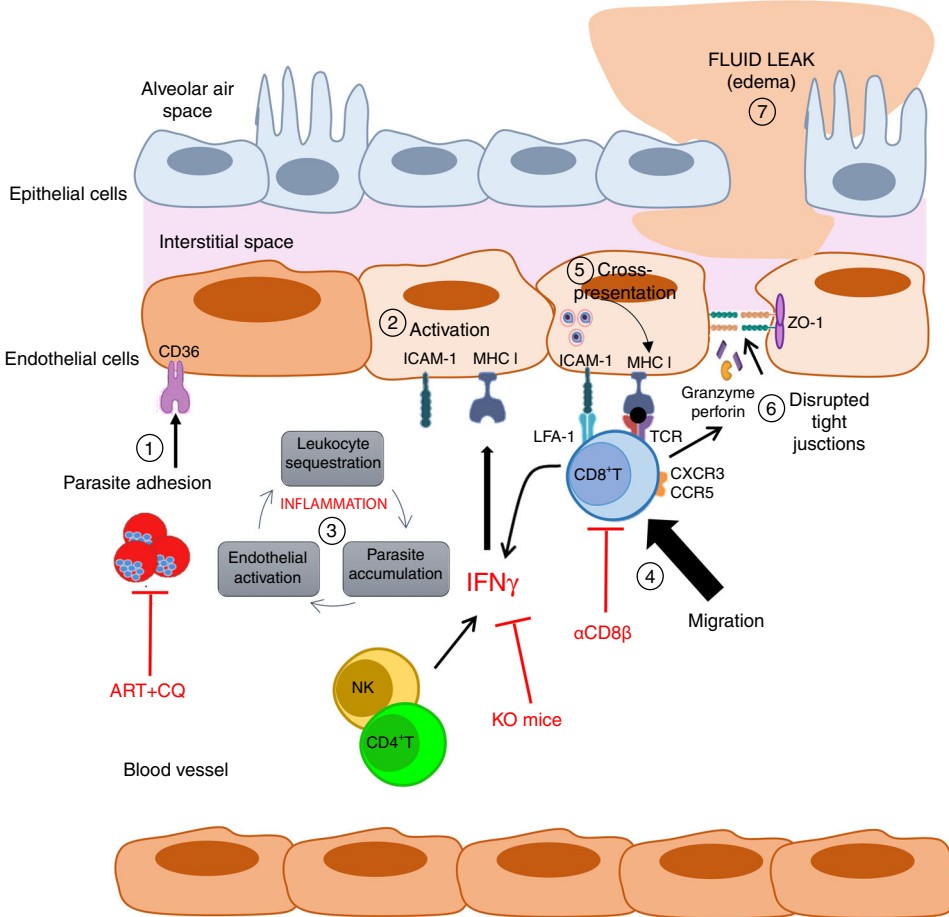

**Fig. 10** Proposed mechanism for malaria-associated lung injury pathogenesis and approaches to modulate pathology. Following PbA*luc* infection in C57BL/6 mice, (1) circulating iRBC (mature stages of the parasite) bind to CD36 receptor expressed on lung endothelial cells, initiating and amplifying the local immune response. (2) Interaction between iRBC and endothelial cells induces the lung endothelial cells to secrete chemokines and cytokines, and adhesion receptors (CD36, ICAM-1), as well as major histocompatibility complex (MHC) class I. (3) These cytokines and chemokines recruit leukocytes (CD4[+] T cells, NK, neutrophils, monocytes) and platelets, that together with iRBC accumulate in the lung microvessel and the activated endothelial cells, start a positive inflammatory feedback loop. Secretion of IFN-γ by CD4[+] T and NK cells induce the activation and proliferation of T cells and increases the expression of adhesion molecules and MHC-I in the lung endothelial cells. (4) Parasite specific-CD8[+] T cells that have been primed in the spleen, migrate into the lungs and bind to endothelial ICAM-1 via LFA-1. (5) IFN-γ-stimulation enhances the antigen processing capability of lung endothelial cells, to present the malaria antigen loaded on MHC-I to specific-CD8[+]T cells. (6) Upon antigen recognition, CD8[+]T cells produce IFN-γ and secrete granzyme and perforin that will disrupt the tight junctions, destabilizing the lung microvascular barrier function, (7) resulting in increased permeability to water (edema). In red, are depicted the approaches used in this study to prevent the pathology: anti-malarial treatment (ART + CQ) and absence of IFN-g that abrogated cross-presentation of malaria antigen by the lung endothelial cells and protected mice from pulmonary vascular leakage; anti-CD8β treatment that protected mice from pulmonary vascular leakage

as the two pathologies are mediated by parasite-specific CD8[+]T cells and parasite antigen cross-presentation, resulting in iRBC sequestration in the tissue. Neurological symptoms and brain edema occur when iRBC accumulation is maximal and when specific CD8[+]T cells migrate to the brain (6–7 dpi)[20,40]. Conversely, lung vascular leakage occurs one day (6 dpi) after maximal accumulation of iRBCs and specific CD8[+]T cell migration to the lungs (Supplementary Fig. 2A). Another study showed that loss of IFN-γ increased CD8[+]T-cell accumulation in the lungs (32, 57) (Fig. 7d) but not the brain[28]. This finding suggests that tissue differences, in terms of the cytokine/chemokine milieu and possibly the presence of other cell types (i.e., DCs) during infection, determine different inflammatory pathways yet lead to the same outcome (i.e., rupture of the endothelial barrier). This hypothesis is supported by the fact that in the PbA model, susceptibility to ALI/ARDS does not always correlate with ECM susceptibility in congenic mice[14]. In human malaria, studies have reported cases of NCM patients with respiratory

involvement[49,50], whereas others that have shown patients with ARDS in conjunction with CM[51,52].

A main clinical complication of MA-ARDS in humans is that it can persist even after treatment, when the parasite is largely eliminated and barely detectable. Some have proposed that this effect is due to a lingering inflammatory response[5,50]. Here, we tested whether a strong combined antimalarial treatment against an established infection could prevent lung injury in mice. We found that when treatment was given on 5 dpi, the parasite load in the lungs decreased and pulmonary vascular leakage was abrogated (Fig. 9b, c). Interestingly, we observed that leukocytes, CD8[+]T cells and Pb1-specific CD8[+]T cells accumulated in the lungs of mice that received the treatment, when the level of peripheral parasitemia was almost near to zero. Nevertheless, we demonstrated that even in the presence of an inflammatory CD8[+]T cell response, absence of parasites prevented antigen presentation by lung endothelial cells, explaining the absence of pulmonary vascular leakage (Figs. 9g and 10). The lung is a major

site of iRBC sequestration during *Plasmodium* infection in humans[53,54], and strategies such as antimalarial treatment associated with adjunct supportive therapies to reduce parasite sequestration may prove beneficial against ALI and ARDS.

Altogether, our data bring insights into the underlying mechanisms of ALI/ARDS development during PbA infection. We found that the interactions between sequestered iRBC, lung endothelial cells and CD8[+]T cells are essential for the development of vascular leakage. The understanding on how the immune response could be modulated in order to prevent the differentiation of CD8[+]T cells into malaria-specific effector cells, and/or to prevent their migration into the lungs, strongly supports the development of approaches that can target this pathogenic pathway to treat malaria-associated ARDS.

## Methods

**Mice**. All mice used in this study were derived from a congenic C57BL/6J background. Male or female C57BL/6, TCRβ[−/−], uGFP, IFN-γ[−/−], and LR-BSL8.4 TCR transgenic mice aged 5–7 weeks from the Biomedical Resource Centre (BRC), Singapore, were housed under the same specific pathogen-free conditions. Mice were group-housed on a 12:12 light/dark cycle with access to food and water.

**BSL8.4 TCR transgenic mice**. BSL8.4 TCR transgenic mice (8.4a[+/−]8.4b[+/−] RAG1[−/−]) expressing *TCRα* and *TCRβ* genes on the RAG1[−/−] background that specifically recognize the Pb1 rodent malaria epitope were created. Briefly, the *Vβ* and *Dβ* segments of BSL8.4 were PCR-assembled with the *Jβ2.1* genomic DNA segment (including 50 bp of J-C intron) and inserted into the XhoI/SacII sites of the pTβcass cassette[55] (provided by the Benoist-Mathis laboratory, Harvard Medical School, USA). After removing the vector backbone by restriction digest, the gene cassette was microinjected into C57BL/6J oocyte. The resulting 8.4b[+/−] line was bred to homozygosity on the RAG1[−/−] background. To generate the 8.4a line, the full-length BSL8.4 *TCRα* cDNA sequence was inserted into the BamHI site of the pES4 vector[56] (provided by Prof. William Heath, University of Melbourne, Melbourne, Australia). The 8.4a[+/−] line was crossed onto the RAG1[−/−] background, but 8.4a[−/−] mice were embryonic lethal. 8.4a[+/−]8.4b[+/−] RAG1[−/−] mice were generated by crossing the resulting 8.4a[+/−]RAG1[−/−] and 8.4b[+/−]RAG1[−/−] lines; 98% of the CD8[+]T cells were positive for the SQLLNAKYL-H-2D[b] (Pb1) tetramer[20] (Supplementary Fig. 4).

**Parasite and infection parameters**. The transgenic *Plasmodium berghei* ANKA (231cl1) line expressing luciferase and GFP under the control of the ef1-α promoter (referred to here as PbA*luc*)[57], was provided by Dr. Christian Engwerda (QIMR, Brisbane, Australia). All mice were infected i.p. with 10[6] parasitized RBCs (iRBCs; prepared in 100 µl (Alsever's buffer) that had been stored in liquid nitrogen[17]. Parasitemia was monitored by flow cytometry daily from 3 to 7 dpi or 7 dpi using anti-CD45 mAb coupled to APC, 5 µg/mL Hoechst 33342 (Sigma) and 8 µM dihydroethidium (Sigma) dyes[58].

**Drug treatment**. Infected mice were administered 0.8 mg chloroquine diphosphate (Sigma-Aldrich) and 0.6 mg artesunate (Guilin No2 Pharmaceutical Factory, China) by i.p. injection on 5 and 6 dpi. Artesunate (dissolved in 0.9% NaCl) and chloroquine (dissolved in distilled water and then filtered) was given twice daily (one dose in the morning and one in the afternoon, 8 h apart)[59].

**CD8[+]T-cell depletion**. Mice were administered one dose of 0.75 mg of rat IgG1 anti-mouse CD8β (clone 53–5.8, BioXCell) antibody, which depletes CD8[+]T cells but not CD8α[+] DCs or CD8α[+] Langerhans cells. Similarly, control (CTR) mice were administered the same dose of 0.75 mg of rat IgG1 isotype control, anti-trinitrophenol (clone TNP6A7, BioXCell) antibody in certain experiments. The antibody was diluted in PBS and was given on 6 dpi by i.p. injection. To determine depletion efficacy, 20 µl whole blood from infected or uninfected mice was collected into 100 µl PBS/10 mM EDTA by tail bleeding before and after antibody treatment. Cells were treated with ACK lysis buffer to remove RBCs, then analysed by flow cytometry with an anti-mouse CD8α (clone 53–6.7) BV605™ antibody (BioLegend, USA). Depletion efficacy was ≥99%.

**Determination of parasite sequestration in the lungs**. Total iRBCs present in the lungs of infected mice was examined in vivo using the IVIS imaging system (Xenogen, Alameda, Cal). As for ex vivo bioluminescence quantification in the lungs, mice were first anesthetized with ketamine (150 mg/kg)/xylazine (10 mg/kg) for 10 min, before i.p. injection of 200 µl D-luciferin potassium salt (Caliper Life sciences) in PBS (5 mg/ml). After 2 min, mice were perfused, and individual lungs were removed and placed in a petri dish. Bioluminescence signals were quantified using the IVIS system[17].

**Determination of pulmonary edema and vascular leakage**. Naïve and infected C57BL/6 mice were euthanized in CO$_2$ chamber and the lungs were removed, placed in pre-weighed Eppendorf tubes and weighed. The lungs were then incubated at 80 °C (using dry water bath) overnight with the Eppendorf tube lid open and the resulting dry lungs were weighed. Pulmonary edema was calculated as the ratio of wet-to-dry lung weight. Tracer-653 dye (Molecular Targeting Technologies), which passively distributes through the blood vessels, was also used to image pulmonary vascular leakage. Here, mice were anesthetized using an isoflurane anesthesia system and 100 µl Tracer-653 (diluted in 1 mL of PBS 1×) was administered via retro-orbital injection and allowed to circulate for 1 h. Fluorescence images were captured using the IVIS system at 640 nm (excitation) and 680 nm (emission) for 0.25–8 s (Supplementary Fig. 1A–C). The data were analysed with Living Imaging 3.0 software to obtain the radiance efficiency [p/s/cm²/sr]/[µW/cm²]. The data are represented as the average radiant efficiency of fluorescence emitted by Tracer-653 dye normalized to naive mice. We are aware that some signals measured in vivo were from the liver. However, the liver has a fenestrated endothelial vasculature that will allow the tracer to diffuse equally in naïve and infected animals. However, since the data were expressed as ratio of infected to naïve, this eliminates the liver background signal. As seen in the supplementary Fig. 1A–C, the fluorescence intensity of Tracer-653 dye from ex vivo measurement of extracted lungs were higher compared to those measured in vivo based on the region of the lungs of naïve and PbA*luc*-infected mice at 7 dpi (Fig. 1d), confirming that we are indeed measuring lung vascular leakage.

**Determination of pulmonary edema by MRI**. MRI of naive (non-infected) and PbA*luc*-infected mice lungs were captured with a Bruker Biospec 94/30 9.4 Tesla scanner (Bruker, Germany) using a T2 weighted RARE imaging sequence with TR/TE = 1200 ms/20 ms. MRI images were taken consecutively from 4–7 dpi. Post processing and quantification of edema were based on manual segmentation/volume calculation using Image J[60].

**Lung immunostaining, tissue clearing, and light sheet imaging**. Mice were anesthetized with ketamine (150 mg/kg)/xylazine (10 mg/kg) for 10 min and intra-cardiac perfusion of PBS was performed at 7 dpi. The lungs were removed, rinsed in DPBS and fixed in 4% PFA (Sigma) overnight at 4 °C. The samples were washed extensively with DPBS and then permeabilized through a graded methanol series (50, 70, 95, and 100%) with a 30 min incubation for each step. The samples were then incubated in 100% methanol/20% DMSO for 1 h before rehydration back to DPBS with 30 min incubation per step. All steps were performed at room temperature. The permeabilized samples were blocked at 37 °C overnight in blocking buffer (2.5% goat serum, 2.5% donkey serum, 5% BSA, 0.2 M glycine, 20%DMSO, 0.3% TritonX-100 in DPBS), washed extensively with DPBS and incubated with primary antibodies in dilution buffer (2.5% goat serum, 2.5% donkey serum, 0.2% BSA, 0.2 M glycine, 5%DMSO, 0.3% TritonX100 in DPBS) over two nights. Primary antibodies used were: chicken anti-GFP (Abcam, ab13970; 1:200 dilution), Armenian hamster anti-CD31 (Merck, 2H8; 1:100 dilution) and rat E-Cadherin (Thermo Fiscer Scientific, ECCD-2; 1:100 dilution). The samples were washed six times for 30 min each at room temperature with 3% NaCl+0.3% TritonX-100 in DPBS, twice with DPBS and incubated with secondary fluorophore-conjugated antibodies (1:200 in dilution buffer) for two nights, washed as described and then taken forward for tissue clearing. All antibody incubations were carried out at 37 °C. Subsequently, lung tissues were cleared by Benzyl Alcohol Benzyl Benzoate (BABB) protocol (Supplementary Fig. 8B). Briefly, samples were dehydrated with graded methanol (as described above), and then equilibrated with BABB: methanol (1:1) for 30 min and the refractive index was matched with BABB (1:2) for 30 min until clear. Cleared lungs in BABB were imaged under a light-sheet Ultramicroscope (LaVision BioTec GmbH, Bielefeld, Germany) fitted with a 2× objective (numerical aperture 0.5), at a 1.0× zoom (to image the lung lobe) or 3.2× zoom (to visualize cellular details).

**Confocal microscopy to assess the pulmonary endothelium**. Cleared lungs were visualized under a FV1000 Olympus Confocal microscope fitted with a 20× long working distance objective, at a 2.0 zoom. Density of tissue parenchyma (tissue compactness parameter) in the 10 µm-thick sections was quantified in Imaris v.9.1.2 (Bitplane) software by creating a surface mask for the CD31 channel and calculating tissue compactness as a ratio of the volume in the mask to the total volume of the selected region of interest.

**Immunofluorescence to assess epithelium damage**. FFPE sections of control, infected and CD8[+]T cells depleted mouse lungs were deparaffinized by incubating in HistoChoice® clearing reagent (Sigma-Aldrich) for 15 min, followed by washing with 100% ethanol for 10 min. The sections were rehydrated by sequentially immersing slides in 90, 70, 50% ethanol, followed by water and antigen retrieval was performed with Target Retrieval Solution, pH 9 (S2367, Dako). Slides were taken for immunofluorescence by blocking with 10% BSA (Sigma-Aldrich) for 45 min, followed by incubating with primary antibodies for 1 h at room temperature. The sections were then washed twice with PBS for 5 min before incubating with secondary antibodies for 45 min at room temperature. Finally, the sections were washed with PBS and mounted in Mowiol mounting media (Sigma).

Following antibodies were used: E-Cadherin (Thermo Fisher Scientific, ECCD2; 1:100), ZO-1 (Thermo Fisher Scientific, 1A12; 1:100) and secondary Alexa-conjugated fluorophore antibody (donkey anti-mouse Alexa 488 and goat-anti-rat Alexa 555) (Life technologies; 1:200). All antibodies were diluted in blocking solution (10% BSA in PBS). Images were acquired at 40× magnification and staining intensity was analysed using Imaris software. Mean intensity of line profiles from 10 randomly selected apical areas of ZO-1 staining were quantified.

**Lung histopathology.** Naive, infected anti-CD8β−treated and untreated C57BL/6 mice were anesthetized with ketamine (150 mg/kg)/xylazine (10 mg/kg) for 10 min before intra-cardiac perfusion with PBS on 7 dpi. Lungs were immediately harvested and fixed in buffered 4% (v/v) formaldehyde for paraffin embedding and H&E staining. Images of H&E-stained histologic lung sections were captured at 40× magnification using an Olympus DP71 camera mounted on an Olympus BX51 microscope. Slides were read and severity of pathological changes was graded on a scale of 1 to 5 by a board-certified pathologist as follows: 0: no abnormalities detected (NAD); G1: minimal; G2: mild; G3: moderate; G4: marked; G5: severe[61]. The grading/score was based on semi quantitative assessment of all lung fields present on all sections on the slide. The severity grades are mainly determined by the extent or an estimate of the percent of tissue involvement with regards to a particular lesion/pathology.

**Lung tissue immune-cell isolation.** Mice were anesthetized with ketamine (150 mg/kg)/xylazine (10 mg/kg) for 10 min and intra-cardiac perfusion of PBS was performed 7 dpi. The lungs were removed, placed in complete RPMI medium (Life Technologies, USA) (containing 10% FBS and 1% penicillin/streptomycin), and cut into small pieces. Subsequently, each lung was digested for 1 h at 37 °C with 50 mg/ml collagenase type 4 (Worthington, USA) and 10 mg/ml DNase I (Roche, Switzerland). Cells were passed through a 70 µm cell strainer, treated with cold ACK lysis buffer, washed and counted with a hemocytometer. The cells were then stained with various antibodies conjugated with fluorochromes (see next section) and fixed in 1% PFA.

**Leukocytes and myeloid staining.** Cells isolated from PBS-perfused lungs at 7 dpi were first stained with Live/Dead Aqua (Life Technologies, USA) and then incubated with 50 µl blocking buffer (1% rat and mouse serum [Sigma-Aldrich] in FACS buffer [0.5% BSA, 2 mM EDTA in PBS]). The fluorochrome or biotin-conjugated monoclonal antibodies specific for mouse leukocytes and myeloid markers used are indicated in Supplementary Table 2. All antibodies were purchased either from BD Biosciences, e-Biosciences or Biolegend. Analysis was carried out by gating on singlets and Aqua−CD45+ cells, followed by the CD45+CD3+ and CD45+CD3− fractions for leukocytes (Supplementary Fig. 1D), the CD45+CD11b+MHCII+CD11c+ fraction to discriminate DCs from monocyte-macrophage cells, and the CD45+MHCII−CD11c+ fraction to define monocytes, neutrophils and macrophages (Supplementary Fig. 1E). Cell samples were acquired using a Fortessa cytometer (Becton Dickinson) and the data were analysed using FlowJo software v.10.0 (Tree Star).

**PbA specific-CD8+T cells tetramer and intracellular staining.** Lung-resident leukocytes were first stained with Live/Dead Aqua (Life Technologies, USA), and then incubated with PE-labeled SQLLNAKYL-H-2Db (Pb1) tetramer for 15 min on ice. The cells were then incubated on ice for 30 min with anti-CD8 mAb coupled to BV605 (Biolegend, USA) and anti-CD16/32 coupled to APC-Cy7 (Biolegend, USA). Next, the cells were washed and fixed in 1% PFA for 20 min. For intracellular staining, cells were incubated with 10 µg/ml Brefeldin A (eBioscience) in 1 ml RPMI media at 37 °C for 3 h. The live cells were determined by staining with Live/Dead Aqua for 30 min followed by surface staining with PE-labeled SQLLNAKYL-H-2Db (Pb1) tetramer for 15 min. The samples were then fixed in 1% PFA overnight at 4 °C. The following day, the cells were permeabilized with 0.5% w/v Saponin (Sigma Aldrich) and intracellular stained (antibodies listed in Supplementary Table 2) on ice. The cells were then washed and re-suspended in FACS buffer before acquisition on a Fortessa cytometer (Becton Dickinson) and analysis with FlowJo software v. 10.0 (Tree Star).

**CyTOF marker labeling and data acquisition.** Cells were isolated from PBS-perfused spleen or lungs a 5 or 7 dpi, respectively, processed, plated and stained in a 96-well U-bottom plate (BD Falcon). The cells were washed once in FACS buffer (4% FBS, 2 mM EDTA, 0.05 % Azide in 1× PBS), and then incubated on ice with 200 µM cisplatin (Sigma-Aldrich) for 5 min. FACS buffer (100 µl) was used to quench the reaction, and the cells were then stained with heavy metal isotope-labeled antibodies on ice. Antibody conjugation was performed[62,63], and after 30 min the cells were washed twice in FACS buffer and once in PBS, and then fixed with 2% PFA in PBS (Electron Microscopy Sciences) at 4 °C overnight. The next day, the cells were washed twice in permeabilization (perm) buffer (BioLegend), and stained with unique, dual-combination heavy metal isotope-labeled antibodies (Supplementary Table 3) barcoded for 30 min on ice. The cells were then washed once with perm buffer and incubated in FACS buffer for 10 min on ice. The cellular DNA was labeled at room temperature using 250 nM Iridium Intercalator (Fluidigm) diluted in 2% PFA/PBS (1:2000), for 20 min. Finally, the cells were washed

twice with FACS buffer and twice with distilled water before resuspending in distilled water. Bromoacetamidobenzyl-EDTA (BABE)-linked and DOTA-maleimide (DM)-linked metal barcodes were prepared[63].The cells were acquired on HELIOS system (Fluidigm).

**Mass cytometry data analysis.** Samples were pre-gated on live, single CD45+CD19−CD90+CD8+T cells. The data were transformed according to Arcsinh (inverse increase sine hyperbolic) with a co-factor of 15. Each sample was then randomly down-sampled to 1161 events (lowest CD8+ T-cell count across all samples). The data then underwent Uniform Manifold Approximation and Projection (UMAP) dimensionality reduction using the umap-learn v2.4.0 Python package, with 1,000 epochs and 15 nearest neighbors[64]. A heatmap was generated using the median summarization of parameter intensities across the CD8+T cells of each sample.

**Adoptive transfer of purified CD8+T cells.** C57BL/6J, µGFP or BSL8.4 TCR transgenic mice were used as infected donors. At 7 dpi, spleens from infected and naïve donors were harvested and after processing, the splenocytes from naïve or infected donors were pooled. CD8+T cells were isolated by negative selection using magnetic beads, according to manufacturer's protocol (mouse CD8α+T cell isolation kit; Miltenyi Biotec), resulting in a population of at least 95% CD8+T cells from the pooled naïve or infected donors splenocytes. The purified cells were adjusted to 3.5 × 10⁶ CD8+T cells in 200 µl, and then retro-orbital injected into PbA*luc*-infected TCRβ−/− mice at 3 dpi. Pulmonary vascular leakage was assessed in vivo in recipient mice after 96 h using Tracer-653 dye. A small proportion of splenocytes from infected and pooled naive donors were stained with Live/Dead Aqua (Life Technologies, USA) followed by mouse monoclonal antibodies against CD3 (clone 17A2), CD4 (clone RM4-5) and CD8 (clone 53-6.7) for 20 min. The purity of the CD8+T cell population was determined on a Fortessa cytometer (BD Biosciences).

**Lungs endothelial cells cross-presentation.** Anaesthetized mice (ketamine/xylazine) were exsanguinated at 7 dpi and the lungs were removed and dissected into single lobes, as described by the manufacturer (Lung Dissociation Kit, MACS-Miltenyi Biotec). The lobes were transferred into gentle MACS C tube containing an enzyme mix (enzyme D and F), and homogenized using a MACS dissociator for 30 min at 37 °C. The cells were re-suspended, passed through a 70 µm cell strainer, and then rinsed with 2.5 ml 1× Buffer S. The cell suspension was then centrifuged at 300 × g for 10 min, the supernatant was discarded and the pellet was treated with ACK lysis buffer and washed. The cells were incubated with mouse monoclonal antibodies diluted in FACS buffer against CD45 (clone 30-F11) conjugated with APC-Cy7 and CD31 (PECAM-1) conjugated with PE-Cy7 for 30 min on ice. Finally, the cells were washed and resuspended in 0.3 ml FACS buffer containing 0.1 µg/ml DAPI. The following DAPI− cells were sorted using a BD FACS ARIA II (Becton Dickinson) cell sorter: CD45+CD31− (leukocytes) and CD45−CD31+ (endothelial cells) (Supplementary Fig. 8A). Next, 100 µl of each sorted cell population was seeded in a 96-well filter plate and 3 × 10⁴ LR-BSL8.4a cells[20] were added to each well and incubated overnight. On the next day, the cells were fixed with 2% formaldehyde and 0.2% glutaraldehyde diluted in water for 5 min, before being stained with 5 mM potassium ferricyanate and ferrocyanate, 2 mM MgCl₂ and 1 mg/ml X-gal diluted in PBS for 6 h at 37 °C. Images of each well were captured on a CTL ImmunoSpot® Analyser, and the blue spots, representing activated LR-BSL8.4a cells, were manually counted. The total number of blue spots was log transformed.

**Lung endothelial cells staining.** The single-cell suspension of sorted lung endothelial cells was first stained with Live/Dead Aqua (Life Technologies, USA), and then with 50 µl blocking buffer containing mouse monoclonal antibodies against H-2Db (KH95, MHCI) conjugated with FITC and IAb/IEb (M5/114.15.2, MHCII) conjugated with efluor 450 (Biolegend).

**Primary culture of lung endothelial cells.** Lung endothelial cells were isolated from naive C57BL/6 mice (as described for the ex vivo cross-presentation assay) and the cell culture was established[65]. Mice were perfused with DMEM, and then the lungs were excised, washed with PBS, cut into 1 × 1 × 1 mm³ pieces and placed in a six-well plate. Tissues were recovered with DMEM-high glucose supplemented with 20% FBS without antibiotics. After 60 h, lung tissues were discarded, and the media was exchanged for fresh DMEM containing 20% FBS and P/S (penicillin/streptomycin), and every 2–3 days thereafter. Primary cultured lung endothelial cells were first stained with Live/Dead aqua (Life Technologies, USA) and then incubated with 50 µl blocking buffer containing the antibodies listed in Supplementary Table 2. Cell samples were acquired using a Fortessa cytometer (Becton Dickinson) and the data were analysed using FlowJo software v.10.0 (Tree Star).

**Lung endothelial cell cross-presentation in vitro.** Lung endothelial cells were stimulated with 10 ng/ml recombinant mouse IFN-γ (R&D Systems). After 24 h, the lung endothelial cells were exposed to 3 × 10⁶ freeze-thawed PbA*luc* schizonts and incubated for a further 24 h. The lung endothelial cells were then washed and

then incubated with $6 \times 10^4$ reporter cells in 0.4 ml RPMI complete medium. These cells were co-incubated overnight before the reporter cells were resuspended and transferred to a 96-well filter plate for X-gal staining.

**Statistical analysis**. The statistical analyses performed depended on the distribution (parametric or nonparametric) of the data. All analyses were performed using Prism 6 (GraphPad Software). Where the data followed a normal distribution (based on Shapiro-Wilk normality test), the Student's $t$-test for two groups or ANOVA with Bonferroni post hoc test for multiple groups was used; otherwise, for $n < 5$, Mann–Whitney or Kruskal–Wallis test was used. Values obtained from parasitemia, imaging and lungs microvessel cross-presentation assays were log-transformed for normalization. A $p$-value $< 0.05$ was considered statistically significant.

**Study approval**. All protocols were approved by the BRC Institutional Animal Care and Use Committee (IACUC #181314) following the National Advisory Committee for Laboratory Animal Research (NACLAR) guidelines and Agri-Food and Veterinary Authority (AVA) rules.

**Reporting summary**. Further information on research design is available in the Nature Research Reporting Summary linked to this article.

## Data availability

The source data underlying Figs. 1a, 2a–d, 6d, h and 7c and Supplementary Figs. 1a and 5d are provided as a Source Data file.

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

## Acknowledgements

The authors would like to thank the Benoist-Mathis laboratory (Harvard Medical School) for providing the pTβcass cassette used to generate the BSL8.4 transgenic mice. We also thank Dr. Anis Larbi and the SIgN Flow Cytometry core for assistance with the sorting, the SIgN Mouse Core for support with mice, and BRC animal Gene Editing Laboratory for oocyte injection. The authors would like to thank Insight Editing London for editing the manuscript prior to submission. SN was supported by a postgraduate scholarship from the Yong Loo Lin School of Medicine, National University of Singapore. This project was funded by Agency for Science, Technology and Research (A*STAR) to a core grant to SIgN. The Flow cytometry core is part of the SIgN Immunomonitoring platform, supported by a BMRC IAF 311006 grant and BMRC transition funds #H16/99/b0/011 (A*STAR).

## Author contributions

Conceived and designed the experiments: C.C., S.N., A.B., L.R. Performed the experiments: C.C., S.N., A.L., J.H., A.B., E.B., B.G., S.V.H., C.B.O., S.H.W.; Analyzed the data: C.C., S.N., J.H., A.B., S.V.H., E.N., J.G., C.B.O., L.R. Wrote the paper: C.C., S.N., L.R. Revised the manuscript: C.C., S.N., A.B., S.H.W., E.B., B.G., S.V.H., C.B.O., E.N., J.G., L.N., L.R.

## Additional information

**Competing interests:** The authors declare no competing interests.

