## [Peer Review File · Nature Communications]

Reviewers' Comments:

Reviewer #1:

Remarks to the Author:

The interesting study by Claser et al. shows that cross-presentation of malarial antigens occurs in the lungs of mice infected with PbA in an IFN γ -dependent way. Furthermore, they show that adoptive transfer of CD8 T cells from infected mice into infected mice devoid of T cells (TCR β ^{-/-}) is sufficient to recapitulate the lung injury. This is very similar to what happens in the brain of these mice (model of experimental cerebral malaria), as previously published by this group in an interesting series of papers. The exploration of these mechanisms in the pulmonary pathology of malaria is an excellent addition to the field.

Major comments

1. The authors state that the damage induced by the CD8 T cells is reversible (e.g. line 361). However, this is not supported by the data. To show that it is reversible, the integrity should be compared before (day 6) and after (e.g. day 7 or day 8) antiCD8 treatment (administered at day 6). Only if the damage is lower after antiCD8 treatment than before, one can conclude it is reversible. The difference observed here (on day 7 with and without antiCD8) only reflects that the disease progression is blocked by the antiCD8 treatment. The idea that perforin or granzyme-mediated cytotoxicity could be involved is also in contradiction with the reversibility, since killing endothelial cells is not something reversible.
2. Along the same line, it is in fact not fully clear whether the cross-presentation by the endothelial cells is important for the pathology. Although the data are highly interesting and suggestive, it cannot be excluded that the cross-presentation might be an epiphenomenon. The adoptive transfer data do show that CD8 T cells are essential, but one cannot fully exclude that the CD8 T cells mediate the pathology through a different mechanism, unrelated to endothelial cross-presentation. This needs to be addressed.
3. Adoptive transfer of CD8⁺ T cells: subpopulations of activated CD8 T cells often express CD11b, which may result in their removal by the negative selection procedure (line 651-653). How does this affect the results?
4. The data analysis and visualisation of CyTOF experiment can probably be improved. The data analysis should also be performed on CD8 T cells from lungs only, to obtain a better identification and visualization of the subpopulations. Are the markers shown in Fig 3 the only ones (out of the 39 included in the CyTOF panel) which are differentially expressed? In the text (line 152), it is stated that the lung CD8 T cells display a fully activated phenotype. Do these activated CD8 T cells form one homogenous population, or can different subpopulations be identified and quantified? The data of the other markers should be shown as well (e.g. in supplementary files). The legend of Fig 3 indicates n = 3. Was this experiment repeated? Are the raw data deposited somewhere?

Minor comments

1. Line 31: replace 'we observe' by 'we confirmed', as this is not a new finding.
2. Line 58-60: This is an old definition in patients, currently no distinction is used anymore between ALI and ARDS in patients. In mice, ALI is still often used as a less severe form of ARDS, but often blood gases are not measured in mice.
3. Line 65: fluid resuscitation is not a therapy of ARDS, it can even be a cause.
4. Line 82: PbK173 is not a good model of ARDS in malaria, as it does not cause alveolar edema, as

shown by Hee et al (no increase in protein in BALF).

5. Line 212-214: decrease in compactness of the tissue: not very clear what this means. Decrease in CD31 expression? Or swollen lungs? The fixation procedure used does not allow to make any conclusion on the swelling of the lungs, as putting the lungs as such in fixative usually results in complete deflation of the lungs.

Reviewer #2:

Remarks to the Author:

In this manuscript the authors explore the mechanisms underlying the lung pathology observed in mice infected with a parasite, *Plasmodium berghei* ANKA (PbA), that causes cerebral malaria (CM) in a well established mouse model. These authors contributed substantially to our understanding of the mechanisms underlying brain pathology in CM, namely PbA-specific CD8+ T cell activated by PbA antigens cross presented on brain endothelium promote blood brain barrier dysfunction, brain swelling and death. In this manuscript the authors provide extensive and convincing evidence that similar mechanisms are central to lung pathology in PbA-infected mice. Lung pathology in PbA-infected mice is a less studied disease as compared to CM although respiratory distress often accompanies CM in *P. falciparum*-infected children. This manuscript appears to provide important evidence that respiratory distress (RD) and CM occur in the same individuals by the same mechanisms in the mouse model. As the mouse model may provide an opportunity to search for therapeutics that would treat both lung and brain disease in children, I would encourage the authors to report on the progression of CM in the mice treated for RD and the relationship between the two.

The survival curve (Fig. 1) shows that most mice die around d10 p.i. but the lung pathology is only reported through d7 p.i. for most parameters. Is lung pathology contributing to death? Are the survival curves similar in the conditions where the lung pathology is blocked (Fig. 4,5). Is CM blocked/reversed under conditions in which lung diseases are improved? The authors describe non-CM (NCM) experimental CM (ECM) mice (Fig. 1F) suggesting that CM contributes to lung pathology and show that lung vascular leakage is greater in the latter. I assume these are all PbA-infected mice but could not find a description of NCM mice. Can the authors comment on this relationship.

Rebuttal comments

Reviewers' comments:

Reviewer #1 (Remarks to the Author):

The interesting study by Claser et al. shows that cross-presentation of malarial antigens occurs in the lungs of mice infected with PbA in an IFN γ -dependent way. Furthermore, they show that adoptive transfer of CD8 T cells from infected mice into infected mice devoid of T cells (TCR β ^{-/-}) is sufficient recapitulate the lung injury. This is very similar to what happens in the brain of these mice (model of experimental cerebral malaria), as previously published by this group in an interesting series of papers. The exploration of these mechanisms in the pulmonary pathology of malaria is an excellent addition to the field.

Major comments

Q1. The authors state that the damage induced by the CD8 T cells is reversible (e.g. line 361). However, this is not supported by the data. To show that it is reversible, the integrity should be compared before (day 6) and after (e.g. day 7 or day 8) antiCD8 treatment (administered at day 6). Only if the damage is lower after antiCD8 treatment than before, one can conclude it is reversible. The difference observed here (on day 7 with and without antiCD8) only reflects that the disease progression is blocked by the antiCD8 treatment. The idea that perforin or granzyme-mediated cytotoxicity could be involved is also in contradiction with the reversibility, since killing endothelial cells is not something reversible.

We agree with the reviewer that we did not demonstrate reversibility based on our data. Therefore we replaced the word “reversible” with “prevented or halted” (line 179 and 351).

On the other hand, we did show that the pulmonary vascular leakage was significantly reduced on day 7 compared to untreated group. It is stated in the text (lines 147-148) that the anti-CD8 β antibody was given on 6 dpi, “when CD8⁺ T cells were already in the lung tissue and vascular leakage was observable”.

Concerning the role of granzyme and perforin, in the discussion we rewrote the sentence by saying that “granzyme B, and possibly perforin, may be involved in the rupture of endothelial cell lining through reduced expression of the tight junction protein” as our data seems to indicate that. This is an agreement with recent finding showing that granzyme B and perforin released by CD8⁺ T cells do kill directly brain endothelial cells but mediate the decrease of expression of tight junction proteins in the brain vasculature during ECM (Nacer et al., 2014; Swanson et al., 2016).

Q2. Along the same line, it is in fact not fully clear whether the cross-presentation by the endothelial cells is important for the pathology. Although the data are highly interesting and suggestive, it cannot be excluded that the cross-presentation might be an epiphenomenon. The adoptive transfer data do show that CD8 T cells are essential, but one cannot fully exclude that the CD8 T cells mediate the pathology through a different mechanism, unrelated to endothelial cross-presentation. This needs to be addressed.

We disagree with the reviewer and we review the evidence presented in the revised manuscript for a central role of lung endothelial cross-presentation. We have addressed this point also in the discussion.

These are the following:

1. As clearly stated by the reviewer, CD8⁺ T cells are the main effector cells. To perform their function, CD8⁺T cells must recognize their target cells through presentation of malaria-derived peptide by MHC class I molecules. We have shown here that lung endothelial cells but no other cells (resident or infiltrated) from infected animals do present malaria antigens to CD8⁺ T cells (using a reporter cell lines expressing the T cell receptor specific for the malaria epitope Pb1).

2. High numbers of CD8⁺T cells (alone) or activated effector CD8⁺ T cells in the lungs were not sufficient for the pathology to occur. This was shown in infected mice deficient for IFN γ or mice that were treated with anti-malarial drugs on 5 dpi. Since CD8⁺ T cells were present in the KO mice and in drug treated mice, if they were acting by attracting or participating in the activation of other cells, we should observe vascular leakage. In fact, we did observed an increased in total leukocytes numbers in IFN γ KO mice (supplemental Figure S6), which suggest that myeloid cells subtype could be increased.

3. As we have demonstrated before for the brain endothelial cells (Howland *et al.*, 2013; Howland *et al.*, 2015), IFN γ is also essential for cross-presentation of lung endothelial cells (in part because it induces MHC class I expression and facilitate antigen presentation). In IFN γ KO mice cross-presentation was equivalent to background levels (Figure 7) despite presence the of CD8⁺ T cells.

4. If cross-presentation is not involved, this implies that CD8⁺ T may attract other cells to the lungs to mediate the pathology. To test this, we quantified the immune cells subtypes infiltrated in the lungs of infected mice depleted or not of CD8⁺ T cells. As observed in the table below (also included as Supplementary table 4, added in the revised manuscript), there is no significant difference in numbers of the different immune cells subtypes between the two groups, except for CD8⁺ T cells. This strongly suggests that migration to the lungs of other cells is independent of CD8⁺ T cells. Moreover, if these cells (monocytes, neutrophils or else) were pathogenic, we should see an increased in lungs vascular leakage on 7dpi since antibody depletion is done on 6dpi (when pathology is already observable).

	Cell number		P Value*
	CTR (n=4)	α CD8 β (n=5)	
Absolute total cell count	4970000 \pm 1454510	4608000 \pm 1639854	NS
CD4	211116 \pm 58513	205738 \pm 47480	a
CD8	162508 \pm 65923	6339 \pm 4613	NS
NK	208976 \pm 49112	150954 \pm 38009	NS
Monocytes	103155 \pm 57587	116658 \pm 112838	NS
Neutrophils	85781 \pm 31695	92317 \pm 29438	NS
cDC	2981 \pm 2290	1450 \pm 248.6	NS
Mono derived DC	37587 \pm 26784	24356 \pm 6320	NS
Macrophages	4816 \pm 1699	9421 \pm 7473	NS
Interst Macro	4985 \pm 4929	4384 \pm 1932	NS
MDM	156700 \pm 69134	269191 \pm 151692	NS
Alveolar macro	79508 \pm 46360	87418 \pm 4288	NS
Definition of abbreviations: N = naïve; cDC = conventional DC; MDM = mono derived macrophages; NS = not significant			

*Significance level of CTR versus α CD8 β on 7 dpi. Cell numbers are indicated in the right-hand end column. Significance levels are indicated as follows: [‡] $P < 0.05$. Mann Whitney test was used.

Q3. Adoptive transfer of CD8+ T cells: subpopulations of activated CD8 T cells often express CD11b, which may result in their removal by the negative selection procedure (line 651-653). How does this affect the results?

A: To address this question, we have run a FACS analysis on the CD8⁺ T cells isolated from the spleen of naïve and PbA-infected mice before and after negative selection. As shown in the figure below, there is no difference in the percentage (%) of CD8⁺LFA-1⁺CD11b⁺ cells before and after negative selection through the column, excluding any possible interference in the data presented.

Figure 1: (A) Gating strategy used to analyse CD8⁺LFA-1⁺CD11b⁺ cells before and after negative selection to isolate CD8⁺T cells from the spleen of naive (n=3) and PbAluc-infected mice (n=3) at 7 dpi. **(B)** Percentage (%) of CD8⁺LFA-1⁺CD11b⁺ cells from the spleen of PbAluc-infected mice (n=3) at 7 dpi, before and after negative selection to isolate CD8⁺T cells. The data represent the mean ± SD.

Q4. The data analysis and visualisation of CyTOF experiment can probably be improved. The data analysis should also be performed on CD8 T cells from lungs only, to obtain a better identification and visualization of the subpopulations. Are the markers shown in Fig 3 the only ones (out of the 39 included in the CyTOF panel) which are differentially expressed? In the text (line 152), it is stated that the lung CD8 T cells display a fully activated phenotype. Do these activated CD8 T cells form one homogenous population, or can different subpopulations be identified and quantified? The data of the other markers should be shown as well (e.g. in supplementary files). The legend of Fig 3 indicates n = 3. Was this experiment repeated? Are the raw data deposited somewhere?

The purpose of this experiment was to identify activation markers differently expressed on CD8⁺ T cells present in the lungs (at 7 dpi) compared to spleen (at 5 dpi considered “priming”

phase). We have followed the reviewer request and the spleen data is now separated from the lungs data and kept as Supplementary figure (Supplementary figure S2). All markers analyzed (see below) were also added in supplementary material (Supplementary fig 2B-C). A re-analysis of the data was performed and as shown below and in figure 3 (revised manuscript), the activated CD8⁺ T cells indeed form one homogenous population. The “n” values represent individual mice. All samples were stained and run in parallel using barcoding to greatly reduce technical variation between samples. Individual replicate UMAP plots (Supplementary Figure 2B) show a high degree of reproducibility between replicate mice. All data has been presented in the paper and therefore not deposited elsewhere.

Supplementary Figure S2. Quantification and characterization of CD8⁺ T cells present in the lungs and spleen. (A) UMAP dimensionality reduction of CD8⁺ T cells isolated from naïve (n=3) and infected spleen (n=3) and lungs (n=3) (5 and 7 dpi, respectively), color-coded by individual samples. The median expression intensities of effector and activation markers probed were plotted and summarized as heat map. (B) Median expression intensities of all phenotypic markers probed across individual lung samples were plotted and summarized as heat map.

Figure 3. UMAP dimensionality reduction of CD8⁺T cells. (A) UMAP dimensionality reduction of lung CD8⁺T cells isolated from naïve (n=3) and *PbAluc*-infected (n=3) mice at 7 dpi was color-coded. A representative plot of three mice is shown. (B) Lung CD8⁺T cell clusters segregation by UMAP identified a total of 15 different clusters. Median expression of lung phenograph clusters across individual samples plotted and summarized as a heat map. (C) UMAP dimensionality reduction of lung CD8⁺T cells color-coded by relative intensity in each channel. The figure shows only the highly expressed markers.

Minor comments

1. Line 31: replace 'we observe' by 'we confirmed', as this is not a new finding.

The word has been replaced

2. Line 58-60: This is an old definition in patients, currently no distinction is used anymore between ALI and ARDS in patients. In mice, ALI is still often used as a less severe form of ARDS, but often blood gases are not measured in mice.

As requested, we only kept the definition that ARDS, the severe form of ALI, which is characterized by alveolar inflammation, alveolar-capillary membrane damaged and pulmonary edema

3. Line 65: fluid resuscitation is not a therapy of ARDS, it can even be a cause.

The “fluid resuscitation” has been removed

4. Line 82: PbK173 is not a good model of ARDS in malaria, as it does not cause alveolar edema, as shown by Hee et al (no increase in protein in BALF).

PbK173 has been removed

5. Line 212-214: decrease in compactness of the tissue: not very clear what this means. Decrease in CD31 expression? Or swollen lungs? The fixation procedure used does not allow to make any conclusion on the swelling of the lungs, as putting the lungs as such in fixative usually results in complete deflation of the lungs.

With the tissue compactness parameter, we are trying to quantitate density of tissue parenchyma. It is qualitatively clear from Fig5B that during infection tissue appears sparse compared to naive condition as revealed by CD31 immunostaining. To quantitate this change, we have simply created a mask for the CD31 signal on several randomly chosen 20um thick sections and computed the volume of this mask. This volume is normalised to the physical dimension of the region of interest. This measurement is independent of intensity of CD31 signal and the parameter is not indicative of tissue swelling or deflation, but is a correlate of infection. We have defined better this parameter in the method section

Reviewer #2 (Remarks to the Author):

In this manuscript the authors explore the mechanisms underlying the lung pathology observed in mice infected with a parasite, *Plasmodium berghei* ANKA (PbA), that causes cerebral malaria (CM) in a well-established mouse model. These authors contributed substantially to our understanding of the mechanisms underlying brain pathology in CM, namely PbA-specific CD8+ T cell activated by PbA antigens cross presented on brain endothelium promote blood brain barrier dysfunction, brain swelling and death. In this manuscript the authors provide extensive and convincing evidence that similar mechanisms are central to lung pathology in PbA-infected mice. Lung pathology in PbA-infected mice is a less studied disease as compared to CM although respiratory distress often accompanies CM in *P. falciparum*-infected children. This manuscript appears to provide important evidence that respiratory distress (RD) and CM occur in the same individuals by the same mechanisms in the mouse model. As the mouse model may provide an opportunity to search for therapeutics that would treat both lung and brain disease in children, I would encourage the authors to report on the progression of CM in the mice treated for RD and the

relationship between the two. The survival curve (Fig. 1) shows that most mice die around d10 p.i. but the lung pathology is only reported through d7 p.i. for most parameters. Is lung pathology contributing to death? Are the survival curves similar in the conditions where the lung pathology is blocked (Fig. 4,5).

We thank the reviewer for her/his comments.

As he/she rightly noted, both ECM and ALI involve the same mechanisms: parasite and CD8⁺ T effector cells sequestration in the tissue and IFN γ –dependent cross-presentation.

In the PbA model, ECM occur only during a window period (from day 6 to 12). This is why we have chosen day 7, when most mice will die of neurological complications (ECM). Any intervention targeting CD8⁺ T cells or reducing parasite load, is preventing both ECM and ALI, although ALI can occur in the absence of ECM (figure 1F). However, there are subtle differences between ALI and ECM. As an example, in IFN γ KO mice, CD8⁺ T cells do not migrate to the brain while they do migrate to the lungs. We are in the process of deciphering the difference in migration of CD8⁺ T cells into the two organs, by looking at the chemokine production profiles. It is our hope to find a different profile, allowing us to prevent ALI or ECM separately.

As requested by the reviewer, we performed an additional experiment (new supplementary figure 3), where we show depletion of α CD8 β extend survival because it prevent both ECM (these mice do not develop neurological signs) and ALI. Mice that survived develop high parasitemia and anemia, and this is known to be one cause of later death.

Supplementary Figure S3. Anti-CD8 β depletion efficiency. (A) Peripheral parasitemia level and (B) survival curve of CTR (n=8) and α CD8 β -treated (n=8) mice. The data represent the means \pm SD; (A) **p<0.01 by Mann-Whitney test. (B) ****p<0.0001 by log-rank (Mantel-Cox) test.

Is CM blocked/reversed under conditions in which lung diseases are improved? The authors describe non-CM (NCM) experimental CM (ECM) mice (Fig. 1F) suggesting that CM contributes to lung pathology and show that lung vascular leakage is greater in the latter. I assume these are all PbA-infected mice but could not find a description of NCM mice. Can the authors comment on this relationship.

1- ECM was not observed in mice where lung disease was improved (CD8 depleted mice, IFN γ KO mice and anti-malarial treated mice).

2- We assessed if both pathologies, ALI and ECM, were linked or not. We have showed that in fact they occur independently. It is stated that ALI occurs independently of neurological

signs (lines 109/110 and 402 - discussion). Depending on the experiment, 60 to 100% of C57BL/6 mice infected with PbA develop ECM, but all develop ALI. In addition, we did not mention that lung vascular leakage was greater in ECM mice (Fig 1F).

3- A sentence providing description of NCM has been added. Line 94/95: "Mice that do not develop ECM are referred as non-ECM (NCM) mice, succumbing to death due to high parasitemia and anemia."

References

Howland, S.W., Poh, C.M., Gun, S.Y., Claser, C., Malleret, B., Shastri, N., Ginhoux, F., Grotenbreg, G.M. & Renia, L. (2013) Brain microvessel cross-presentation is a hallmark of experimental cerebral malaria. *EMBO molecular medicine*, **5**, 984-999.

Howland, S.W., Poh, C.M. & Renia, L. (2015) Activated Brain Endothelial Cells Cross-Present Malaria Antigen. *PLoS pathogens*, **11**, e1004963.

Nacer, A., Movila, A., Sohet, F., Girgis, N.M., Gundra, U.M., Loke, P., Daneman, R. & Frevert, U. (2014) Experimental cerebral malaria pathogenesis--hemodynamics at the blood brain barrier. *PLoS pathogens*, **10**, e1004528.

Swanson, P.A., 2nd, Hart, G.T., Russo, M.V., Nayak, D., Yazew, T., Pena, M., Khan, S.M., Janse, C.J., Pierce, S.K. & McGavern, D.B. (2016) CD8+ T Cells Induce Fatal Brainstem Pathology during Cerebral Malaria via Luminal Antigen-Specific Engagement of Brain Vasculature. *PLoS pathogens*, **12**, e1006022.

Reviewers' Comments:

Reviewer #1:

Remarks to the Author:

The authors have adequately addressed most of the comments. However, some inaccuracies have remained or were added with the new data. These issues need to be addressed.

1. Suppl Table 4. This table should be checked thoroughly. The data for CD8 T cells appear erratic, since after the depletion their number should be significantly decreased. Presumably the significance of CD4 and CD8 was switched.
2. Issue of CD8 T cell purification for the adoptive transfer (previous comment N°3). The authors provide data in their answer to this comment in their rebuttal letter. From these data, it clearly appears that only naive cells seem to remain after the purification, as most LFA1+CD8+ cells have disappeared. In naive mice: 15.7% LFA+ before, only 0.96% after negative selection. In infected: 46.1% LFA1+ before, only 4.91 after negative selection. This clearly shows that the negative selection removes most of the activated CD8+ T cells. Furthermore, on line 668 the authors mention that 95% of CD8 T cell purity is achieved, which is chiefly incorrect since the figure shows that only 92% (naïve) and 77% (infected) purity was obtained. This has to be corrected. How does this affect the results? The limitations of the purification in this experiment must be discussed in the paper, and the authors are encouraged to show this figure (FACS data of the purification) in the supplemental data. This will not affect the quality of the paper but rather enhance the correct and transparent reporting.
3. Supplementary Fig S2 panel B. Insufficiently clear. What are the numbers (left hand side of the figure)? If these are the sample numbers, indicate which ones are infected, spleen, lung etc.
4. Figure 3. Panel A: For a more easy understanding: provide a legend what the red cells and the grey cells represent. Panel B: why can't the markers be shown instead of the numbers (left on the heatmap)?
5. Figure 4, panels H and I: the pictures clearly show bronchial epithelium. However, in the case of malarial ARDS, showing the lung parenchyma (alveoli) would be more appropriate. The authors are encouraged to replace the pictures by other pictures (different fields from the same sections), showing alveoli rather than bronchial epithelium. Presumably this does not affect the conclusions, but would be more supportive of the ZO1 quantifications in relation to the malaria ARDS pathology.

Reviewer #2:

Remarks to the Author:

The authors have adequately addressed my comments. I believe this manuscript is novel and will be of interest to both immunologists and infectious disease biologists.

REVIEWERS' COMMENTS:

Reviewer #1 (Remarks to the Author):

The authors have adequately addressed most of the comments. However, some inaccuracies have remained or were added with the new data. These issues need to be addressed.

1. Suppl Table 4. This table should be checked thoroughly. The data for CD8 T cells appear erratic, since after the depletion their number should be significantly decreased. Presumably the significance of CD4 and CD8 was switched.

We thanks the reviewer for pointing out our mistake. The significance was corrected.

2. Issue of CD8 T cell purification for the adoptive transfer (previous comment N°3). The authors provide data in their answer to this comment in their rebuttal letter. From these data, it clearly appears that only naive cells seem to remain after the purification, as most LFA1+CD8+ cells have disappeared. In naive mice: 15.7% LFA+ before, only 0.96% after negative selection. In infected: 46.1% LFA1+ before, only 4.91 after negative selection. This clearly shows that the negative selection removes most of the activated CD8+ T cells.

Although the reviewer brought up an important point that the majority of activated CD8⁺T cells were removed after purification, our adoptive transfer experiment clearly shows that even in the absence of these cells, we were able to recapitulate the vascular leakage.

Furthermore, on line 668 the authors mention that 95% of CD8 T cell purity is achieved, which is chiefly incorrect since the figure shows that only 92% (naïve) and 77% (infected) purity was obtained. This has to be corrected. How does this affect the results? The limitations of the purification in this experiment must be discussed in the paper, and the authors are encouraged to show this figure (FACS data of the purification) in the supplemental data. This will not affect the quality of the paper but rather enhance the correct and transparent reporting.

The 77% purity was obtained when the experiment was performed to answer the rebuttal question. On line 668 we mentioned that ~95% of purity was achieved because in the 3 experiments performed for the paper, the purity was 94.9, 97.9 and 98.6%.

In the supplementary figure 5B (see below), we are already representing the flow cytometry dot plot showing the purity before and after enrichment using CD8 α ⁺T cell isolation kit. In this figure, we can see that after enrichment we were able to obtain 98.6% of CD8⁺T cells.

3. Supplementary Fig S2 panel B. Insufficiently clear. What are the numbers (left hand side of the figure)? If these are the sample numbers, indicate which ones are infected, spleen, lung etc.

We apologize that Supplementary Figure 2 did not appear clear. The loss of resolution could have been caused during the file upload, because in the original PDF the images are clear (see below).

4. Figure 3. Panel A: For a more easy understanding: provide a legend what the red cells and the grey cells represent. Panel B: why can't the markers be shown instead of the numbers (left on the heatmap)?

It is already stated in the legend that panel A is a UMAP plot representation. We have added a line in the figure legend explaining that “the red dots represent where the CD8⁺T cells are located”.

Regarding panel B, the heat map represents the different clusters found in the lungs (clearly stated in the legend), not the markers. The different markers are shown in supplementary figure 2D.

5. Figure 4, panels H and I: the pictures clearly show bronchial epithelium. However, in the case of malarial ARDS, showing the lung parenchyma (alveoli) would be more appropriate. The authors are encouraged to replace the pictures by other pictures (different fields from the same sections), showing alveoli rather than bronchial epithelium. Presumably this does not affect the conclusions but would be more supportive of the ZO1 quantifications in relation to the malaria ARDS pathology.

The images in Figure 4H-I were provided to demonstrate the overall effect of the infection on the lung. In addition, the lung parenchyma 3D imaging provided in figure 5B (lung tissue compactness) also recapitulates the effect of the parasite not only in the alveoli but also on the whole tissue. The lung tissue compactness was quantified on cleared tissue (see image below – added as supplementary figure 8B), a 3D image (see also supplementary video) that allow us to look at a thicker volume

compared to 5 μm tissue sections (figure 4H-I). Figures 4H-I, where FFPE IHC was performed, the idea was to look at the integrity of the tissue at molecular level. The e-cad and ZO-1 staining of bronchi reveal that the integrity is compromised upon infection.